

# Cloud, precipitation and radiation responses to large perturbations in global dimethyl sulfide

Sonya L. Fiddes[1,2,3], Matthew T. Woodhouse[3], Zebedee Nicholls[1,4], Todd P. Lane[2], and Robyn Schofield[2]

[1]Australian-German Climate and Energy College, University of Melbourne, Parkville, 3010 Australia
[2]ARC Centre of Excellence for Climate System Science, School of Earth Sciences, University of Melbourne, Parkville, 3010, Australia
[3]Climate Science Centre, Oceans and Atmosphere, Commonwealth Scientific and Industrial Research Organisation, Aspendale, 3195, Australia
[4]ARC Centre of Excellence for Climate Extremes, School of Earth Sciences, University of Melbourne, Parkville, 3010, Australia

*Correspondence to:* S. Fiddes
(sonya.fiddes@climate-energy-college.org)

**Abstract.** Natural aerosol emission represents one of the largest uncertainties in our understanding of the climate system. Sulfur emitted by marine organisms, as dimethyl sulfide (DMS), constitutes one fifth of the global sulfur budget and yet the distribution, fluxes and fate of DMS remain poorly constrained. In this study we quantify the role of DMS in the chemistry-climate system and determine the climate's response to large DMS perturbations. By removing all marine DMS in the Australian Community Climate and Earth System Simulator (ACCESS) - United Kingdom Chemistry and Aerosol (UKCA), we find a top of atmosphere radiative effect of $1.7 \, \mathrm{Wm^{-2}}$. The largest responses to removing marine DMS are in stratiform cloud decks in the Southern Hemisphere's eastern ocean basins. These regions show significant differences in low-cloud (-9%), radiation ($+7 \, \mathrm{Wm^{-2}}$ in short wave incoming surface radiation) and large-scale rainfall (+15%) when all DMS is removed. We demonstrate a precipitation suppression effect of DMS-derived aerosol in stratiform cloud deck regions, coupled with an increase in low cloud fraction. The increase in low cloud fraction is an example of the aerosol lifetime effect. Other areas of low cloud formation, such as the Southern Ocean and stratiform cloud decks in the Northern Hemisphere, have a relatively weak response to DMS perturbations. Our study highlights the need for further modelling and field studies of natural aerosols and their impact on cloud and precipitation, in particular in Southern Hemisphere stratiform cloud regions.

## 1 Introduction

Current understanding of the global climate is underpinned by the concept of the radiation budget; the balance of energy entering and leaving the Earth's atmosphere. Aerosols play an important role in this budget, having direct (McCormick and Ludwig, 1967) and indirect effects via cloud processes (Twomey, 1974; Albrecht, 1989). Aerosols produce a net cooling effect at the surface, with the total aerosol effective radiative forcing estimated as $-0.9 \, \mathrm{W \, m^{-2}}$ by the most recent Intergovernmental Panel on Climate Change (IPCC) report, substantially offsetting the effect of well mixed greenhouse gases effective radiative forcing of $2.8 \, \mathrm{W \, m^{-2}}$ (Myhre et al., 2013). However, large uncertainty in aerosol radiative forcing remains ($\pm 0.5 \, \mathrm{W \, m^{-2}}$ in



the 2013 IPCC report), and is in fact the largest source of uncertainty to the overall radiation budget for the current climate (Myhre et al., 2013; Carslaw et al., 2013). Uncertainties due to aerosols affect not only the radiation budget, but also chemical and meteorological parameters such as ozone concentration and photolysis (Kushta et al., 2014), cloud formation, albedo, temperature and precipitation (Seinfeld et al., 2016; Rotstayn et al., 2015; Rosenfeld et al., 2014).

Natural aerosol sources account for the largest portion of this uncertainty, explaining up to 45% of the variance of aerosol forcing, compared to 34% from anthropogenic aerosol emissions (Carslaw et al., 2013). Second to volcanic activity, DMS produced by marine organisms makes up approximately 19% of global sulfur emissions, producing a DMS flux ($flux_{DMS}$) of 17.6 $Tg\ year^{-1}$ (Sheng et al., 2015), though estimates range from 15-32 $Tg\ year^{-1}$ (Elliott, 2009; Woodhouse et al., 2010). DMS concentrations and fluxes remain poorly constrained by observations and in climate models (Hopkins et al., 2016), and
its role in the climate system is subject to debate (Charlson et al., 1987; Quinn and Bates, 2011).

       Charlson, Lovelock, Andreae and Warren (CLAW) proposed a hypothesis by which marine organisms, primarily phytoplankton, regulate their environment via the increased production of dimethyl sulfonium propionate (DMSP) when stressed, for example due to warm sea surface temperatures (SSTs) (Charlson et al., 1987). DMSP is degraded via bacterial processes to DMS in the ocean (Yoch, 2002), some of which is vented into the atmosphere (Charlson et al., 1987). Once in the atmo-
sphere DMS has a lifetime of 1-2 days (Kloster et al., 2007; Korhonen et al., 2008), and oxidises to form sulfuric acid and ultimately contribute to the aerosol burden. This additional source of aerosol can directly or indirectly influence the radiation budget and potentially cool local SSTs (although this has not been proven in the literature), hence reducing phytoplankton stress. The DMS-climate system is summarised in Fig. 1. Current understanding of the DMS-climate system implies that no bio-regulatory feedback exists as proposed by the CLAW hypothesis (Quinn and Bates, 2011; Woodhouse et al., 2013). How-
ever, observations show that seasonal cloud condensation nuclei (CCN) variability cannot be explained without a contribution from DMS (Korhonen et al., 2008; Vallina et al., 2006), implying DMS is important for the longer term climate. Complicating this problem is our poor understanding of the global distribution of DMS, ultimately relying on the collection of observations collated and interpolated by Lana et al. (2011), which may considerably misrepresent local DMS concentrations such as over coral reefs (Hopkins et al., 2016) and at the poles (Mungall et al., 2016; Kim et al., 2017).

The relationship of DMS with large-scale climate has been highlighted by numerous global modelling studies. Mahajan et al. (2015) (using the Lana et al. (2011) DMS climatology) and Thomas et al. (2010) (using the Kettle et al. (1999) DMS climatology) found DMS to have a radiative effect of -1.79 $W\ m^{-2}$ and -2.03 $W\ m^{-2}$ at the top of the atmosphere (TOA) respectively. Thomas et al. (2011) doubled DMS, finding a TOA radiative effect of -3.42 $W\ m^{-2}$. These studies removed DMS from the climate system in a one year simulation to quantify its importance on climate and noted the largest changes in
radiation and cloud microphysics in the Southern Ocean and southern Pacific and Indian Oceans.

       Other studies have explored the impact of anthropogenic climate change on marine DMS production, often with opposing conclusions, making it unclear whether marine DMS production would increase or decrease with warming temperatures (for example Bopp et al., 2004; Gabric et al., 2004; Kloster et al., 2007; Cameron-Smith et al., 2011). More recent laboratory experiments have confirmed that under anthropogenic climate change, and in particular ocean acidification, marine organisms
produce significantly less DMS (Hopkins et al., 2011). A global modelling study by Six et al. (2013) found DMS emissions





were reduced by 17% by the end of the century, primarily due to decreasing ocean pH. Six et al. also explored the resultant feedback of these changes in DMS on the climate, as outlined by Fig. 1. The reduced DMS flux was found to cause an additional 0.23-0.48 °C of warming by the end of the century (Six et al., 2013). Reduced DMS flux due to ocean acidification is also found by Schwinger et al. (2017), who found under the Representative Concentration Pathway (RCP) 8.5 to the year

2200, DMS production decreases by 48% assuming a strong sensitivity of DMS production to changes in pH. The authors calculated a DMS temperature sensitivity of -0.041 K per Tg year$^{-1}$ of sulfur.

Conversely, Grandey and Wang (2015) attempted to determine if a significant artificial increase of marine DMS production (due to, for example, ocean fertilisation) in the oceanic ecosystem could offset future warming trends. Under such a scenario of drastically increased atmospheric Flux$_{DMS}$, Grandey and Wang found global temperature increases due to anthropogenic

climate change under RCP 4.5 were partially offset, primarily due to low-mid level cloud feedbacks resulting in a change to the radiative effect of -2.0 W m$^{-2}$. Regional changes in precipitation (both increases and decreases) were also noted, up to as much as 30%.

The interactions between DMS-derived sulfur, its oxidation products and the atmosphere can be highly non-linear, vary regionally and have far-reaching impacts on multiple processes in the climate system (Thomas et al., 2011). These interactions

merit further study and may have important implications in addition to the more commonly reported effects of DMS on global temperature and radiation, for example its influence on cloud and precipitation.

This study has two aims, the first of which is to assess the suitability of the ACCESS (Australian Community Climate and Earth System Simulator) -UKCA (United Kingdom Chemistry and Aerosol) model for examining the role of DMS in the Earth's climate in terms of low, medium and high cloud cover, outgoing TOA shortwave (SW) and longwave (LW) radiation,

incoming surface SW radiation, and precipitation. Secondly, ACCESS-UKCA is used to test the large-scale sensitivity of the climate to prescribed changes in surface water DMS concentrations, DMS$_w$. Two simulations are performed to explore the chemical, aerosol and meteorological implications of DMS$_w$ perturbations. In the first simulation, all DMS$_w$ is removed from the system to determine its current contribution to the climate. In the second experiment, DMS$_w$ is significantly increased, and the results are compared to that of the work by Grandey and Wang (2015).

This paper is organised as follows: Section 2 outlines the methodology used in this study, Section 3 evaluates how well ACCESS-UKCA performs with respect to certain satellite products, Section 4 the sensitivity of the ACCESS-UKCA climate to large perturbations in DMS is analysed and Section 5 provides some discussion and concluding remarks.

## 2   Methods

### 2.1   Model description and set-up

#### 2.1.1   ACCESS-UKCA

The ACCESS-UKCA coupled climate-chemistry model is a platform from which the influences of DMS on the large-scale climate can be evaluated. The physical atmosphere in the ACCESS model is the United Kingdom Met Office's Unified Model



(UM). In this case, UM version 8.4 is used, in conjunction with the UKCA chemistry model (Abraham et al., 2012), which includes the GLObal Model of Aerosol Processes (GLOMAP)-mode aerosol scheme described in Section 2.1.2.

Horizontal grid resolution is 1.25°latitude x 1.85°longitude, with 85 vertical levels, where the model top is located at 85 km. Anthropogenic emissions are prescribed pre-2000 by the Atmospheric Chemistry and Climate Model Intercomparison Project

(ACCMIP) (Lamarque et al., 2010), and post-2000 by the Representative Concentration Pathway 6.0 (van Vuuren et al., 2011). Biomass burning emissions are from the GFED4s database (van der Werf et al., 2017). Emissions of other species required by ACCESS-UKCA, and their original sources, including biogenic emissions, chemical precursors and primary aerosol, are described in detail in Woodhouse et al. (2015). DMS emissions are calculated within UKCA, and are described in Section 2.1.3. Monthly mean SST and sea ice coverage are prescribed as per the Atmospheric Model Intercomparison Project (AMIP)

(Taylor et al., 2015). UKCA is coupled to the ACCESS radiation scheme via $O_3$, $CH_4$, $N_2O$ and aerosol (direct scattering and absorption). Aerosols further influence the large-scale cloud and precipitation schemes via the cloud droplet nuclei (CDN) concentration, allowing changes in the the chemical/aerosol fields to affect the meteorology.

For this study, ACCESS-UKCA is run for the years 2000-2009, with a one year spin-up, 1999. The simulations are nudged to ERA-Interim (Dee et al., 2011), using the horizontal wind component and potential temperature, at 6 hourly intervals in the free

troposphere. Whilst performing a nudged simulation limits the ability of the model to respond to the changes in sulfate aerosol, a free running simulation would not allow us to separate direct aerosol responses from the subsequent process feedbacks that may occur. In this theoretical study, we are interested in the direct impacts of sulfate aerosol on the climate, including the atmospheric composition as well as its effects on radiation, cloud and precipitation.

### 2.1.2    GLOMAP

The GLOMAP-mode aerosol scheme uses two-moment pseudo-modal aerosol dynamics to simulate aerosol size distributions (Mann et al., 2010, 2012). GLOMAP-mode simulates particle compositions with sulfate, sea-salt, elemental and organic carbon in internally-mixed modes (Mann et al., 2010). Dust is treated outside of GLOMAP-mode according to the scheme of Woodward (2001). Processes simulated within GLOMAP-mode include primary emissions, new particle formation, particle growth by coagulation, condensation and cloud processing and removal by dry deposition, and in-cloud and below-cloud scav-

enging (Mann et al., 2010). New particle formation occurs via two mechanisms in ACCESS-UKCA; free tropospheric binary homogeneous nucleation (Kulmala et al., 1998) and organic-mediated boundary layer nucleation (Metzger et al., 2010). The aerosol size distribution is represented in four soluble modes (corresponding to nucleation, Aitken, accumulation and coarse size modes) and one insoluble mode (Aitken). A full description of the scheme can be found in Mann et al. (2010).

### 2.1.3    $DMS_w$ climatology and flux parameterization

The number of $DMS_w$ observations have increased dramatically over the last three decades (Kettle et al., 1999; Lana et al., 2011), although significant gaps, both spatially and temporally, remain. Lana et al. (2011) use observations to derive a gridded $DMS_w$ climatology, used in this study and shown in Fig. 2a. The Lana et al. (2011) climatology shows that high latitude regions





have the highest $DMS_w$ concentrations. Significant sampling biases exist with approximately half of observations collected in late spring - summer, and more than two thirds of the data in the Northern Hemisphere (Lana et al., 2011).

The $flux_{DMS}$ from the ocean to the atmosphere remains poorly parameterised, has large variability in space and time and cannot easily be measured. This subsequently causes large uncertainties in the $flux_{DMS}$ parameterisation, a summary of which

can be found in Elliott (2009). The Liss and Merlivat (1986) $DMS_{flux}$ parameterisation is used here. In this method, $DMS_{flux}$ is given by Eq. 1 where $k$, the piston velocity, is parameterised under three wind induced sea surface regimes: smooth (Eq. 2) and rough (Eq. 3) gas transfer, and a wave breaking/bubble-bursting regime (Eq. 4).

$$DMS_{flux} = k(DMS_w - \frac{DMS_a}{\alpha}) = k(DMS_w \alpha - DMS_a) \tag{1}$$

For $w_{10} < 3.6\ ms^{-1}$:

$$k = 0.17 w_{10} (\frac{SC_{DMS}}{600})^{\frac{2}{3}} \tag{2}$$

For $3.6\ ms^{-1} < w_{10} < 13\ ms^{-1}$:

$$k = 2.85(w_{10} - 3.6)(\frac{SC_{DMS}}{600})^{\frac{1}{2}} + 0.612(\frac{SC_{DMS}}{600})^{\frac{2}{3}} \tag{3}$$

For $w_{10} > 13\ ms^{-1}$:

$$k = 5.9(w_{10} - 13)(\frac{SC_{DMS}}{600})^{\frac{1}{2}} + 26.79(w_{10} - 3.6)(\frac{SC_{DMS}}{600})^{\frac{1}{2}} + 0.612(\frac{SC_{DMS}}{600})^{\frac{2}{3}} \tag{4}$$

Where $DMS_w$ solubility $\alpha$ = 11.4 at 26 °C, $w_{10}$ = 10 m wind speed ($ms^{-1}$) and the Schmidt Number of DMS $SC_{DMS}$, a measure of viscosity/diffusion and a function of sea surface temperature, determined following the method of Saltzman et al. (1993). The denominator in this function is the Schmidt Number of $CO_2$ in fresh water at 20 °C, $SC_{CO_2}$ = 600, which is used to normalise the numerator ($SC_{DMS}$). We assume that the concentration of $DMS_a$ is negligible, as it is orders of magnitude

smaller than that of sea-water.

## 2.2 Model evaluation

In order to provide a climatological evaluation of ACCESS-UKCA, and to put the sensitivity testing of DMS into a real-world context, a comparison to observational data sets is presented. Global means at the surface are calculated over the 2000-2009 period, except for the cloud climatologies which were only available from 2006-2009.

The following global data sets were compared to the model output: low, medium and high cloud fractions from the GCM-Oriented Cloud-Aerosol Lidar and Infrared Pathfinder Satellite Observation Cloud Product (CALIPSO-GOCCP) (Chepfer et al., 2010), radiation fluxes from the Clouds and the Earth's Radiant Energy System (CERES) -Energy Balanced and Filled (EBAF) -TOA Edition 4.0 (Loeb et al., 2009) and CERES-EBAF-Surface Edition 4.0 (Kato et al., 2013) and precipitation from the Tropical Rainfall Measuring Mission (TRMM) (Huffman et al., 2007). Cloud fraction is defined in line with the



CALIPSO-GOCCP: high between 50-44hPa, medium 440-680 hPa and low 680-1000hPa. Direct comparison of cloud fractions between model output and satellites cannot take into account satellite measurement biases, which can be resolved using a cloud satellite simulator such as the Cloud Feedback Model Intercomparison Program (CFMIP) Observation Simulator Package (COSP). COSP was not available in the version of ACCESS-UKCA used here, limiting the comparison. Nevertheless, a useful comparison is still possible.

### 2.3 DMS sensitivity testing

To explore the sensitivity of the global climate to large perturbations in $DMS_w$ concentrations, two experimental simulations were performed and compared to the control run (Ctl). As described above, the Ctl simulation used the Lana et al. (2011) $DMS_w$ climatology, which is shown in Fig. 2a.

In Experiment 1 (Exp.1), $DMS_w$ was set to zero, leaving a flux of 0.72 Tg year$^{-1}$ of sulfur derived from terrestrial sources (for example Jardine et al., 2015). From this we can attribute what role ocean-derived DMS plays in shaping our current climate and enhance our understanding of how the physical processes underpinning the DMS-climate system operate. This may further aid our understanding of how natural aerosols interact with the global radiation budget.

In Experiment 2 (Exp.2), the $DMS_w$ field was set to each latitude's (at the model resolution of $1.25°$) monthly zonal maximum value, following a similar method to Grandey and Wang (2015), and shown in Fig. 2b. This simulation allows further exploration of the physical processes by which DMS can influence the climate, when the perturbations are exaggerated.

Three regions of interest are defined. The Australian region: $45°S$-$10°S$, $110°E$-$160°E$, the Southern Ocean (SO): ocean grid points south of $40°S$ and the South Eastern Pacfic (SEP) that represents an area of significant stratiform cloud decks: $240°E$-$270°E$, $25°S$-$0°$

### 2.4 Global energy budget

Due to the nudging of the model to ERA-Interim and the prescribed SSTs, a direct estimate of how global temperatures might respond to DMS perturbations is not possible. For this reason, a simple energy balance model has been used to estimate the effects of the DMS perturbations on global mean temperatures: the climate component of the Finite Amplitude Impulse Response (FAIR) model. This model is based on that first proposed by Boucher and Reddy (2008) and subsequently used in the most recent IPCC for equivalent emission metric calculations (Myhre et al., 2013). FAIR's climate component is a simple impulse response model which emulates the behaviour of more complex Earth System Models. It is ideally suited to estimating global mean temperature changes resulting from TOA radiative imbalances.

For each experimental run, the radiative effect ($R_{DMS}$) due to increasing or decreasing $DMS_w$ is defined as the difference between the TOA energy balance (Q*) of an experimental run from the Ctl, which can be taken directly from ACCESS-UKCA. By providing this radiative effect to FAIR's climate component, we can estimate the difference in temperature expected across the ten year period under zero $DMS_w$ or enhanced $DMS_w$ conditions, without the need for an ensemble of free-running climate simulations.



## 3   Model evaluation

This section compares selected ACCESS-UKCA fields to satellite-derived observations. In order to give context to this evaluation, the ACCESS-UKCA output is also compared to that of the CMIP5 (Coupled Model Intercomparison Project Phase 5) global climate models (GCMs).

### 3.1   Cloud fraction

The cloud fraction comparison is performed for the years 2006-2009, aligning with the availability of CALIPSO-GOCCP data. ACCESS-UKCA simulates too little low cloud fraction (Fig. 3a-c) over the the majority of the globe (mean bias of -0.16), which is consistent with findings for the CMIP5 GCMs (Cesana and Chepfer, 2012; Nam et al., 2012; Klein et al., 2013). Areas of large stratiform cloud decks in eastern ocean basins are significantly underestimated, by a fraction of >0.5, consistent with other CMIP5 and CFMIP Phase 1 and 2 findings (Bony and Dufresne, 2005; Cesana and Chepfer, 2012; Klein et al., 2013). These low level marine clouds have an important impact on the global radiation budget (Leon et al., 2009) and have been identified as the primary source of uncertainty in tropical cloud-climate feedbacks (e.g. the effects of the cloud albedo) in GCMs (Bony and Dufresne, 2005). These biases have been attributed to poor vertical distribution of clouds in the models, difficulty capturing overlapping cloud layers, the misrepresentation of cloud structures, deficiencies with the statistical parameterisation of clouds and likely problems in the cloud microphysics (Nam et al., 2012). Low clouds over the polar regions and some areas of northern Asia and America are slightly overestimated. The ACCESS-UKCA low-cloud biases over the Arctic are within the range of biases found for the CMIP5 GCMs studied in Cesana and Chepfer (2012). It should be noted that satellite observations are subject to biases in detecting low clouds, particularly over the Southern Ocean.

ACCESS-UKCA reproduces medium cloud fraction (Fig. 3d-f) reasonably well, within ±0.1 in most regions (global mean bias of -0.01). The largest discrepancies are overestimated medium cloud fraction over the Southern Ocean and Antarctica, where the simulated medium cloud fraction is at its highest globally. The Antarctic bias is of opposite sign to the CMIP5 models compared in Cesana and Chepfer (2012). Bodas-Salcedo et al. (2014) note that issues within GCMs around distinguishing between clouds with tops at actual mid level and low level clouds contribute to such biases. The biases in high cloud fractions (Fig. 3g-i) show similar spatial patterns to that of the low cloud fraction, where an underestimate occurs over most of the tropics and mid-latitude. The global mean bias is 0.05. The largest negative biases, of up to 0.3 occurs over the Maritime Continent. Moderate overestimation is noted over the polar regions. These biases are within the range of those found for the CMIP5 models studied in Cesana and Chepfer (2012).

Interestingly, Nam et al. (2012) noted that due to underestimated low clouds in the tropics, the CMIP5 models over compensated by producing optically thick and too bright low clouds and more high clouds, impacting the radiation budget. Here, an underestimation of low clouds is also found, although there is no evidence of an over-compensation of high clouds. Predominantly a small underestimation of high cloud fraction is found in this simulation at tropical to mid latitudes (Fig. 3i).





## 3.2 Radiation

The remaining analyses consider means over the period of 2000-2009. The comparison of the observed and simulated TOA outgoing LW radiation is shown in Fig. 4a-c. The observed global mean of $239.7\,\mathrm{W\ m^{-2}}$ is closely matched by the simulated $241.0\,\mathrm{W\ m^{-2}}$. Compared to the CMIP5 ensemble, which tends to underestimate TOA outgoing LW radiation, $238.6\,\mathrm{W\ m^{-2}}$ from Stephens et al. (2012) and $238.9\,\mathrm{W\ m^{-2}}$ from Wang and Su (2013), TOA outgoing LW radiation in ACCESS-UKCA is slightly overestimated. The regions with the largest biases (both positive and negative) occur in regions of deep convection (Fig. 4c), and align well spatially with the biases in high cloud fractions shown in Fig. 3c. Underestimation by $-3\,\mathrm{W\ m^{-2}}$ of TOA outgoing LW radiation occurs over the polar oceans, which may partly be explained by an overestimation of cloud fraction at all levels, and especially the mid-level clouds (Fig. 3f) in this region.

Spatial biases in the TOA outgoing SW radiation (Fig. 4d-f) are of greater magnitude than that of the LW radiation. In most regions the sign of the outgoing SW radiation bias is opposite to that of the LW radiation. The same processes as described above that block LW radiation from escaping the atmosphere prevent SW radiation reaching the surface, instead reflecting more sunlight thus enhancing the albedo. Globally, ACCESS-UKCA performs reasonably well, simulating the global mean TOA outgoing SW radiation of $101.8\,\mathrm{W\ m^{-2}}$ compared to the observed $99.6\,\mathrm{W\ m^{-2}}$, consistent with the multi-model mean of GCM ensembles from previous studies (Stephens et al., 2012; Wang and Su, 2013). In Fig. 4f, an abrupt change in sign of TOA outgoing SW radiation at $60°\mathrm{S}$ is found, which is also present in the CFMIP comparisons (Bodas-Salcedo et al., 2014). In the Southern Ocean, wrongly assigned mid-level cloud types have been found to be a leading cause of the model underestimation of TOA outgoing radiation (Bodas-Salcedo et al., 2014). In addition, poor representation of the physical processes surrounding super-cooled liquid water in the Southern ocean has been found to account for 27-38% of the total reflected solar radiation (Bodas-Salcedo et al., 2016). Over the Antarctic ice sheets, both TOA outgoing and surface incoming SW radiation are overestimated, due to an underestimation of low clouds allowing the high albedo to reflect too much incoming SW radiation back out to space.

Globally, ACCESS-UKCA overestimates incoming surface SW radiation (Fig. 4g-i), with $202.4\,\mathrm{W\ m^{-2}}$ compared to observations of $198.3\,\mathrm{W\ m^{-2}}$. This overestimation is slightly greater, though within the uncertainty of that found for CMIP5 GCMs of $2\pm6\,\mathrm{W\ m^{-2}}$ (Stephens et al., 2012). Nevertheless, large regional biases of over $\pm30\,\mathrm{W\ m^{-2}}$ exist. The most notable features, apart from those discussed above, are too much incoming SW radiation over the continents and the tropical regions, which can be attributed in part to the underestimated cloud cover. The northern Pacific and Atlantic Oceans, the Arctic Ocean and parts of the Southern Ocean all receive too little incoming SW radiation, consistent with overestimated cloud cover.

## 3.3 Precipitation

Precipitation in ACCESS-UKCA has large positive biases (Fig. 5d-f). Over regions to the north and south of the equator, regions that receive the most annual rainfall, ACCESS-UKCA performs with the least skill, overestimating precipitation by over $2000\,\mathrm{mm\ year^{-1}}$. Poor performance of GCMs in this region are not unusual however (Stephens et al., 2010), with the current CMIP5 GCM ensemble overestimating precipitation in a similar region by more than $1000\,\mathrm{mm\ year^{-1}}$ (Flato et al.,





2013). Stephens et al. (2010) found that models in these regions produce light rain too frequently, indicating that convective processes are poorly simulated. Two of Australia's CMIP5 GCMs, ACCESS 1.0 and 1.3, both overestimate precipitation in this region by similar amounts to that of the ACCESS-UKCA model (Bi et al., 2013). If biases of precipitation are considered as a percentage (not shown), the largest differences (positive) occur in the eastern basins of the South Pacific and South Atlantic
Oceans.

## 4   DMS perturbations

This section aims to quantify the role of DMS in the large-scale climate system. Two experimental simulations have been performed, described in Section 2.3 and Table 1, which involve the removal of all $DMS_w$ (Exp.1) and setting the $DMS_w$ to the zonal maximum (Exp.2).

### 4.1   Exp.1: Zero $DMS_w$

#### 4.1.1   Chemistry response

The 2000-2009 annual mean ocean $flux_{DMS}$ from ACCESS-UKCA is $17.41\,Tg\,year^{-1}$ of sulfur, resulting in an atmospheric DMS ($DMS_a$) annual mean surface concentration of 81.9 ppt. Taking all marine DMS out of the model (but retaining the terrestrial source of $0.72\,Tg\,year^{-1}$ of sulfur) results in a 94% reduction in $DMS_a$ at the surface; throughout the troposphere,
it results in a 98% reduction of $DMS_a$.

The impact of this reduced $flux_{DMS}$ on atmospheric sulfur can be seen in Fig. 6a-b, d-e. Globally, there is a net decrease of 15% of $SO_2$ at the surface. The largest absolute differences are in the tropics and mid-latitudes. Large relative decreases in $SO_2$ occur in the SO and SEP, 84% and 94% respectively. Fig. 7a shows the vertical profile of $SO_2$ for the Australian region (ref), the SO (blue) and the SEP (green). The large peak in concentration at approximately $500\,m$ occurring in the Australian profile
is attributable to industrial and energy-related emissions of $SO_2$, representing lofting by chimneys and smokestacks. The $SO_2$ in Exp.1 is consistently lower than that of the Ctl throughout the troposphere, though for the regional means, the difference begins to decrease closer to the tropopause.

Surface H2SO4 (Fig. 6d-e) shows significant loss in predominantly clean marine areas; the SO has a 79% decrease and the SEP an 84% decrease, compared to a 49% global mean decrease. Interestingly, heavily polluted regions, especially busy
shipping lanes, undergo an increase in $H_2SO_4$. $H_2SO_4$ is a precursor gas, which can participate in new particle formation forming secondary sulfate aerosol, or it can condense onto pre-existing particles. The increased H2SO4 concentration in heavily polluted regions results from a decreased condensational sink (not shown). Similar non-linearities have been described in Thomas et al. (2011).

The vertical profiles of $H_2SO_4$ in Fig. 7b, show that the largest differences between the Exp.1 and the Ctl occur in the free
troposphere (between $1\text{-}10\,km$) for all regions. In all three regions (each considered a clean atmospheric environment), net decreases of $H_2SO_4$ occur.





### 4.1.2 Aerosol response

The majority of gaseous $H_2SO_4$ is taken up by aerosol formation (99.99%) as opposed to being removed by dry deposition (0.01%) (Mann et al., 2010). The peak in nucleation mode number density in the free troposphere in Fig. 7c coincides with the peak concentration of $H_2SO_4$. Surface global nucleation mode number concentration decreases by 9% between Exp.1 and the Ctl (see Fig. 6g-h). Whilst in absolute terms, clean terrestrial regions have the largest decreases, the Australian region only has a relative decrease of 18% in nucleation mode particles. Over the oceans, although few nucleation mode particles exist, there are large relative differences of both signs.

In absolute terms, the differences in the aerosol number concentration are greatest in the smaller aerosol modes, particularly the nucleation mode described above. Fig. 7d-f show the number concentrations for the Aitken mode, accumulation mode, and coarse mode (global maps not shown). The Aitken mode (Fig. 7d) shows some differences between the two simulations, with profiles reflecting reduced new particle formation in the free troposphere and reduced condensation-growth of $H_2SO_4$ onto pre-existing particles in the boundary layer. The largest differences are seen over the Australian region. Similar boundary layer differences are also present in the accumulation mode, with the differences between the Exp.1 and Ctl consistent below 1 km (Fig. 7e). Little difference is seen in the coarse mode throughout the troposphere (Fig. 7f), which in marine regions is dominated by sea salt.

As aerosols grow towards the larger end of the Aitken mode, they become relevant to cloud processes. Fig. 8a shows the Ctl's $N_3$ (condensation) number concentration ($N_3$ signifies particles with a dry diameter greater than 3 nm). The difference in surface $N_3$ number concentration between the Ctl and Exp.1 shows the largest relative decreases occur in clean, coastal regions, predominantly in the Southern Hemisphere, as well as some regions of the SO. In heavily polluted terrestrial regions a small increase in the $N_3$ number concentration occurs. A decrease of 8% is found globally. For the Australian region (representative of a clean, terrestrial region), a decrease of 17% is found. The largest absolute differences are in clean terrestrial regions. Over the SO a relative decrease of 39% occurs at the surface. The SO and the SEP have far fewer aerosols in all modes except the coarse mode (see Fig. 7c-f), where sea salt dominates. This decrease in number concentration in small aerosol modes represents a large portion of the aerosol loading in the region. The increase in nucleation mode particles is reflected in the $N_3$ for the SEP region, via a more moderate decrease of 20%.

Fig. 8d-e show the number concentration of cloud condensation nuclei with dry diameters greater than 70 nm ($CCN_{70}$) for the Ctl and the differences resulting from Exp.1. The largest absolute differences are in the tropics, which, similarly to the $N_3$, have the highest concentration. Relatively, there is a global decrease of 5%, whilst over the Australian region a 7% decrease, the SO an 8% decrease and the SEP a 20% decrease. Differences in cloud droplet number (CDN) are shown in Fig. 8g-h. The relative differences in CDN (Exp.1-Ctl) show a similar spatial pattern to that of the CCN. Global mean CDN decreases by 5%, whilst a decrease of 5% is also found for the Australian region, 8% for the SO and 18% for the SEP. In both the $CCN_{70}$ and CDN, the marine Southern Hemisphere mid latitudes have the largest decreases of 14% (averaged between 5-35°S) despite the SO having some of the larger decreases in $SO_2$ and $H_2SO_4$.





The larger differences in concentration of both CCN and CDN in the oceanic Southern Hemisphere tropics-mid latitudes, compared to the SO, warrant further investigation of how sulfate aerosols are interacting with their background environments, for example cloud processes and pre-existing aerosols. The SO has large concentrations of sea salt particles, which like more polluted regions of the Northern Hemisphere, may provide a buffering effect to reduced DMS-derived aerosols. Additionally, in

areas of persistent low cloud formation, in-cloud aqueous sulfate oxidation is the dominant reaction (over gaseous nucleation), which allows almost instantaneous condensational growth of existing aerosols, and is temperature dependent. We speculate that poor representation of low clouds in the SO may be having further impacts on atmospheric composition modelling than currently realised. A cloud resolving modelling study may be better suited to gain understanding of the complex system described here.

Following the method of Woodhouse et al. (2010, 2013), global and hemispheric sensitivities of CCN to $\text{flux}_{\text{DMS}}$ have been calculated (Table 2). The results presented here a suggest lower CCN sensitivity to $\text{flux}_{\text{DMS}}$ compared to the Woodhouse et al. (2013) study where absolute sensitivities of 94 and $63\,\text{cm}^{-3}/\text{mg}\,\text{m}^{-2}\,\text{day}^{-1}$ of sulfur were found globally for June and December respectively. Similar CCN sensitivities are reported in the Woodhouse et al. (2010) study. The lower sensitivities in our study are likely the result of the large (near 100%) changes in $\text{flux}_{\text{DMS}}$ (the denominator). Relative CCN sensitivities

calculated here compare well with the Woodhouse et al. (2010) and Woodhouse et al. (2015) studies. For example Woodhouse et al. (2010) finds mean hemispheric relative CCN sensitivities of 0.02 for the Northern Hemisphere and 0.07 for the Southern Hemisphere. These results highlight the greater relative importance of DMS in the Southern Hemisphere.

### 4.1.3 Meteorological response

Meteorological responses to the DMS perturbations must be considered carefully. As detailed in the methods section, the

ACCESS-UKCA simulations are nudged to ERA-Interim potential temperature and horizontal winds, preserving synoptic scale meteorology and limiting any feedbacks. Whilst performing a non-nudged simulation would allow the meteorology to respond to changes in the chemistry and aerosol more freely, it would make comparison of the aerosol and meteorological responses more difficult. Within ACCESS-UKCA, GLOMAP-mode is directly coupled to the large-scale cloud and precipitation schemes via the CDN (Abraham et al., 2012), as well as the radiation scheme via aerosols and some (see Section 2.1.1). Convective

rainfall and cloud formation are not directly coupled to the aerosol scheme, but can be indirectly influenced via changes in radiation (which can act on temperature, moisture, etc).

Differences in low cloud fraction occur predominantly in areas with large stratiform cloud decks (Fig. 9a), in the eastern basins of the Southern Hemisphere's oceans and slightly removed from the coastline. The SEP region shows an annual mean decrease in low cloud fraction of 9%. In the Northern Hemisphere (including the north eastern Pacific where significant strat-

iform cloud decks are found) and the SO (where persistent low cloud formation occurs) only small differences are evident, which may in part be due to the modest differences in CCN and CDN concentrations discussed in Section 4.1.2. Stratiform cloud deck fractions are consistently underestimated by ACCESS-UKCA and other GCMs (see Section 3 in comparison to other areas of significant low cloud formation such as the SO. The mechanism behind the different responses (between the SO and cloud deck regions), and whether the long standing model biases exacerbate this requires further investigation.



The decrease in low cloud fraction and aerosol number concentration discussed above lead to an increase in surface incoming SW radiation (Fig. 9c). This increase in surface SW radiation is highest in the regions of stratiform cloud deck formation. In the SEP region there is a mean increase of $7\,\mathrm{W}\,\mathrm{m}^{-2}$.

Decreases of total liquid water ($Q_{cl}$) at 1700 m height shown in Fig. 10a-b) are found in the stratiform cloud deck regions.
Little difference in $Q_{cl}$ occurs at the surface. The decrease in $Q_{cl}$ is coincident with increases in large-scale precipitation in the stratiform cloud decks, regions with very little precipitation (Fig. 10d-e). In the SEP region large-scale rainfall increases by $17\,\mathrm{mm}\,\mathrm{year}^{-1}$ (15%) over the Ctl mean of $111\,\mathrm{mm}\,\mathrm{year}^{-1}$.

In the southern hemisphere stratiform cloud decks, and in particular the SEP region, the model demonstrates a distinct cloud lifetime effect in response to removing DMS in Exp.1. Decreased CDN concentration and the associated increase in cloud
droplet size and increased liquid water lead to increased autoconversion and large scale rainfall. The overall impact is to reduce low cloud fraction.

Figure 10g-h shows the differences in convective rainfall. Whilst the convective rainfall scheme is not coupled directly to GLOMAP-mode, there are differences between the simulations. Convective rainfall decreases in Exp.1 compared to Ctl along the Intertropical Convergence Zone (ITCZ) (a mean difference of $11\,\mathrm{mm}\,\mathrm{year}^{-1}$ between 20°S-20°N). This difference
represents a small fraction (less than 1%) of the total convective rainfall. Relatively, (not shown) the largest differences (a 5% decrease in the SEP) are found once again in eastern basins of Southern Hemisphere stratiform cloud decks.

Seifert and Beheng (2006) note that even when convection schemes are coupled to an aerosol scheme, the effects of CCN on convection, and the resultant precipitation and associated maximum updrafts, differs significantly depending on the cell type and size, making these effects difficult to quantify. Large differences in convective rainfall would not be expected in these
results, due to the meteorological nudging used in the experiments.

## 4.2  Exp.2: Zonally increased $DMS_w$

This section considers the response to zonally enhanced $DMS_w$, resulting in a $flux_{DMS}$ of 37.05 Tg year$^{-1}$ of sulfur (relative to 17.4 Tg year$^{-1}$ of sulfur in the Ctl simulation). For comparison the Grandey and Wang (2015) study used a zonally enhanced $flux_{DMS}$ of 46.1 Tg year$^{-1}$ of sulfur (up from 18.2 Tg year$^{-1}$ of sulfur), under global warming scenarios. Many of the
differences resulting from zonally enhancing $DMS_w$ show similar spatial patterns, with similar magnitude, but reversed sign, compared to Exp.1.

Globally, the differences in $SO_2$ (Fig. 6c) are of comparable magnitude to Exp.1. Increased $SO_2$ concentrations occur over the Australian region, the SO and the SEP: 42%, 172%, 89% respectively. There is a net decrease in $H_2SO_4$ of 14% in Australia, and a larger decrease over the tropical oceans. Over the SO there is an increase of 9%, whilst in the SEP a decrease of 37%.
Similar non-linearities are discussed in terms of doubled DMS in the Thomas et al. (2011) study. These differences in $SO_2$ and $H_2SO_4$ are also clear in the vertical profiles shown in Fig. 7a-b.

Differences in the aerosol modes (see Fig. 7c-f) are of similar magnitude but opposite sign to those noted in Section 4.1.2. Global mean $N_3$, $CCN_{70}$ and CDN increases by 6%, 4% and 5% (Fig. 8c,f,i). Larger differences are seen over the SO of 27%, 15% and 13% and SEP of 14%, 19% and 17% for $N_3$, $CCN_{70}$ and CDN respectively.





Globally, there is little difference in low cloud fraction or $Q_{cl}$, though increases are noted in regions of large stratiform cloud decks (Fig. 9d), showing similar spatial patterns to that of Exp.1. Surface incoming SW radiation has a global mean decrease of -1.75 W m$^{-2}$. This decrease is comparable to the Grandey and Wang (2015) finding of -2.2 W m$^{-2}$ (noting the larger DMS perturbation by Grandey and Wang (2015)). Lastly, decreases in large-scale precipitation are found, again in regions of

stratiform cloud decks (Fig. 10f), whilst general increases in convective precipitation over the tropical oceans occur (Fig. 10i). The Grandey and Wang (2015) study, analysed under a warming climate, also found large relative decreases in precipitation rate predominantly in eastern ocean basins. We find, under the current climate, the largest relative increase in total precipitation (not shown) in the south east basins of the Pacific and Atlantic ocean, however these results presented are much nosier than the Grandey and Wang (2015) results. Whilst Grandey and Wang (2015) find that artificial enhancement of DMS may offset

global warming, which is supported by this study as implied by the decreases in incoming SW radiation at the surface, the precipitation responses warrant further study.

### 4.3   Temperature response

The global 2000-2009 mean of the TOA radiation budget (Q*), and its main components are provided in Table 3. Due to the nudging used in the simulations, we do not expect the TOA Q* to be balanced (i.e. Q* = 0). The differences in Q* seen in

Exp.1 and Exp.2, 1.69 and -1.48 W m$^{-2}$ respectively, show a substantial radiative effect of DMS on the energy budget. The Q* response found for Exp.1 is consistent with the Mahajan et al. (2015) findings of 1.79 W m$^{-2}$. Using the FAIR model's climate component, the 2000-2009 mean temperature response is calculated to be 0.45 °K for Exp.1 and -0.38 °K for Exp.2.

Although other studies generally consider DMS changes under global warming, we can make comparison via the sensitivity of the estimated global temperature response to changes in the flux$_{DMS}$ (see Table 4). In this study, we find a response of

0.027 K per Tg year$^{-1}$ of sulfur in Exp.1, and 0.019 K per Tg year$^{-1}$ of sulfur in Exp.2. These results are of similar magnitude to the Grandey and Wang (2015) study, and in the range of the lowest impact scenario of Six et al. (2013). The other scenarios in the Six et al. (2013) study suggest much higher temperature sensitivities to changes in flux$_{DMS}$, as does the Schwinger et al. (2017) study.

### 5   Discussion and Conclusions

We have used a global climate model, ACCESS-UKCA, which includes a detailed microphysical aerosol module, to conduct sensitivity experiments to determine the role of DMS on several aspects of the climate system.

Evaluation using satellite-derived observations shows ACCESS-UKCA is comparable to other GCMs in the assessed climatological fields. Low, medium and high cloud fractions, compared to the CALIPSO-GOCCP satellite climatology show similar biases to that of the CMIP5 ensemble (Bony and Dufresne, 2005; Cesana and Chepfer, 2012; Nam et al., 2012; Klein et al.,

2013), as do the the radiation fields (Stephens et al., 2012; Wang and Su, 2013). Of particular interest to this study is the model underestimation of large stratiform cloud decks located in the eastern mid-latitude basins of the Earth's oceans. These regions of extensive low cloud produce little rainfall (that is overestimated by the model) and are often regions of high primary



productivity. These biases have not been attributed to a single cause (rather a multitude of theories, as discussed in Section 3.1), indicating a gap in understanding of atmospheric processes in these regions (Nam et al., 2012).

The use of nudging does not allow aerosol and cloud responses to perturbed DMS to affect synoptic scale meteorology, hence the results here represent instantaneous responses in the climate system. Nudging was deemed desirable for this study
to limit computational expense, allowing single runs of 10 years, rather than ensemble runs of greater duration. Other considerations include the inherent uncertainties associated with all climate simulations, including emissions uncertainties (both natural and anthropogenic), and parameterizations and physical representation of atmospheric processes. Nevertheless, where clear shortcomings have been found in comparison to the satellite-derived observations, the ACCESS-UKCA model has been found to perform with comparable skill to current CMIP5 GCMs. Additionally, it is important to note biases in the satellite
products themselves, for example in cloud fraction retrievals as noted in Mace and Zhang (2014) or Protat et al. (2014).

Removing marine $DMS_w$ from the climate system leads to significant responses in chemistry and aerosol concentrations. The differences are largest in $SO_2$ and $H_2SO_4$, with differences becoming smaller through the latter stages of the DMS-aerosol-cloud system (Figure 1). Areas of stratiform cloud decks are particularly sensitive to DMS emissions. On average globally, marine DMS contributes up to 16% of $SO_2$, and 48% of $H_2SO_4$. The Australian region, considered a clean, terres-
trial environment, undergoes greater decreases in these fields than the global mean, whilst the SO and the SEP clean marine environments experience even larger decreases. Marine DMS removal results in an 8% decrease of the $N_3$ number density and a 5% decrease of CCN and CDN globally. In less pristine environments, close to coastlines and in the Northern Hemisphere, a weak response is found in $N_3$, CCN and CDN. Similarly, a weak response is also found in the SO where sea salt makes a large contribution to aerosol.

Changes in meteorological parameters (low cloud fraction, large-scale precipitation, moisture) are largest in regions where large stratiform cloud decks occur, generally in the Southern Hemisphere. In the SEP region we find a decrease of 9% in low cloud formation and a 15% increase in large scale rainfall when DMS is removed. These stratiform cloud decks are regions where considerable model bias exists in low cloud fraction and precipitation. Global mean differences were small, implying a higher sensitivity of these stratiform regions to changes in DMS. We find that DMS in these stratiform regions
plays an important role in cloud processes and precipitation suppression (as discussed in Thomas et al. (2011) or with regards to anthropogenic pollution in Ackerman et al. (2004)). In other regions of significant low cloud formation, aerosol sources such as sea salt and anthropogenic aerosols may buffer the regions from changes in DMS-derived aerosols.

Significant differences in the global radiation budget result from removing marine $DMS_w$ and the subsequent decrease in sulfate aerosol and low cloud fractions. As a result of the decreases in low cloud fraction in the SEP region, there is an increase
of $7\,\mathrm{Wm^{-2}}$ in incoming surface SW radiation. We find removal of $DMS_w$ leads to a global increase in the TOA radiation of $1.69\,\mathrm{Wm^{-2}}$, resulting in an increase in global temperature of approximately $0.45\,^\circ\mathrm{K}$.

Previous studies examining the role of DMS in the climate system have not identified stratiform cloud decks as regions of particular importance, rather highlighting the SO Thomas et al. (2010); Mahajan et al. (2015). Mahajan et al. (2015) estimated the TOA radiative effect of DMS to be $1.79\,\mathrm{Wm^{-2}}$, which is consistent with our results ($1.69\,\mathrm{Wm^{-2}}$), but slightly lower than
the of $2.03\,\mathrm{Wm^{-2}}$ estimated by Thomas et al. (2010) (who used the previous Kettle et al. (1999) $DMS_w$ climatology).



In addition to impacts on radiation budget and temperature, as previously reported in the literature, we are able to quantify the role of DMS in suppressing large-scale precipitation in stratiform cloud deck regions. For example in the SEP region we find an increase in large-scale rainfall of 15% when $DMS_w$ is removed. Furthermore we have demonstrated that marine $DMS_w$ is directly responsible for increasing low cloud fraction in stratiform cloud deck regions, a demonstration of the second aerosol

indirect (or lifetime) effect (Albrecht (1989)).

The zonally enhanced $DMS_w$ simulation responds in almost the opposite manner, with the same regions of the globe showing differences of similar magnitude but opposite sign. The Grandey and Wang (2015) study, with a $flux_{DMS}$ of 44.1 Tg year$^{-1}$ of sulfur estimated a TOA radiative effect of up to -1.28 Wm$^{-2}$ in low-medium cloud regions for the year 2000, and -1.49 Wm$^{-2}$ by 2080. Our study, with a $flux_{DMS}$ of 37.0 Tg year$^{-1}$ of sulfur, finds a global mean radiative effect of -1.48 Wm$^{-2}$ (2000-

2009 mean). Areas of stratiform cloud decks show differences of up to -5 Wm$^{-2}$ of incoming SW radiation. In this study, the estimated global mean temperature change with increased $DMS_w$ is -0.38 °C (2000-2009 mean). The changes in the aerosol and chemistry noted here imply that significant enhancements of DMS may have greater implications for some regions than previously identified in the Grandey and Wang (2015) study.

The estimated temperature responses per unit change in DMS-derived sulfur flux are lower than those reported in the Six

et al. (2013) and Schwinger et al. (2017) studies by approximately a factor of two. The temperature response sensitivities calculated here are comparable to those given in Grandey and Wang (2015). The cause of the discrepancy between the results presented here and those in Six et al. (2013) and Schwinger et al. (2017) is difficult to speculate upon without further informa-tion. However, the discrepancy between these results suggest the need for better observational constraints, and highlights the complexity of the DMS-aerosol-cloud system.

Natural aerosols account for a significant source of uncertainty in climate modelling and radiation budgets (Carslaw et al., 2013). Our study uses the Lana et al. (2011) $DMS_w$ climatology with the Liss and Merlivat (1986) flux parameterization. Whilst this dataset and method are commonplace for DMS-climate studies, both are limited by sparse observations and biases. For example: recent studies have indicated that coral reefs are an as of yet unaccounted for source of marine DMS (Hopkins et al., 2016; Swan et al., 2017; Jones et al., 2017). Furthermore, larger concentrations and/or fluxes of DMS than what we

currently consider have also been found at the poles, especially around sea ice and polynyas (Nomura et al., 2011; Jarnikova and Tortell, 2016; Mungall et al., 2016; Kim et al., 2017).

Observational deficiencies become particularly relevant when considering stratiform cloud deck regions. In the Lana et al. (2011) dataset, the SEP region contains only two ship campaigns collecting measurements in January and February. The cloud deck in the Southern Hemisphere eastern basin of the Indian Ocean has no $DMS_w$ observations. The higher susceptibility

of cloud and precipitation to changes in DMS in these regions suggest that they should be a priority for future atmospheric composition observation campaigns. Providing further motivation for study in these areas, Six et al. (2013); Schwinger et al. (2017) predict an increase in $flux_{DMS}$ in these stratiform cloud regions (despite a global mean decrease) due to climate change by the end of the 21st century.

Global perturbations of $DMS_w$ in this study indicate that significant changes in meteorology are restricted to key marine

boundary layer cloud regions despite global responses in chemistry and aerosol fields. This suggests that whilst inclusion of





updated $DMS_w$ concentrations may have an impact on regional chemistry and aerosol fields, the impact on regional climate may be small, and in terms of global climate, insignificant. This hypothesis will be tested in future studies. For example, a follow-up to this study will examine the role of coral reef-derived DMS in regional and global climate.

To place the conclusions of this study into a broader perspective, we must consider the DMS-climate system within the context of anthropogenic climate change. Hopkins et al. (2011); Six et al. (2013) and Schwinger et al. (2017) have demonstrated that marine phytoplankton is vulnerable to ocean acidification, amongst other oceanic changes expected with global warming, for example impacts on nutrient availability, salinity, mixed layer depths, and light penetration (Kloster et al., 2007). Whilst both the Six et al. (2013) and Schwinger et al. (2017) temperature responses are much larger than found here, our results imply a 25% decrease in $flux_{DMS}$ would result in an increase of $0.1\,^\circ C$ globally. Considering the current Paris Agreement target of limiting global warming to $1.5$-$2.0\,^\circ K$, the sensitivity of ocean-derived sulfate aerosol to warming temperatures and ocean acidification becomes important. Strategies to mitigate anthropogenic climate change must consider not only the effect of increased $CO_2$ on temperatures, but also on ocean pH. Mitigating only temperature increases, e.g. via solar radiation management, may have a short term cooling effects, however without removing $CO_2$ from the atmosphere, ocean acidification will continue to impact on marine life, and as demonstrated here, the climate.

*Author contributions.* SF completed the ACCESS-UKCA simulations, analysis and the initial draft of this manuscript. MW developed the initial model setup and provided advice as to the specific setup requirements of this study. MW helped guide the analysis and contributed significantly to the revisions of this manuscript. ZN provided the FAIR analysis in this manuscript and contributed to the revisions of this manuscript. RS and TL provided advice and guidance on the direction of this study and contributed to the revisions of this manuscript.

*Competing interests.* The authors declare that they have no conflict of interest.

*Acknowledgements.* SLF would like to thank P. J. Rayner and his research group for their helpful discussions. SLF, RS, TPL are supported by the Australian Research Council (ARC) Centre of Excellence for Climate System Science (CE110001028). ZN is supported by the Australian Research Council (ARC) Centre of Excellence for Climate Extremes (CE170100023). SLF, RS, were supported by the ARC Discovery Project: Great Barrier Reef as a significant source of climatically relevant aerosol particles (DP150101649). MTW is supported by the Earth System and Climate Change Hub of the Australian Government's National Environmental Science Programme (NESP). This research was undertaken with the assistance of resources and services from the National Computational Infrastructure (Project q90), which is supported by the Australian Government. SF and ZN are supported by the Australian Government Research Training Program Scholarship.





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





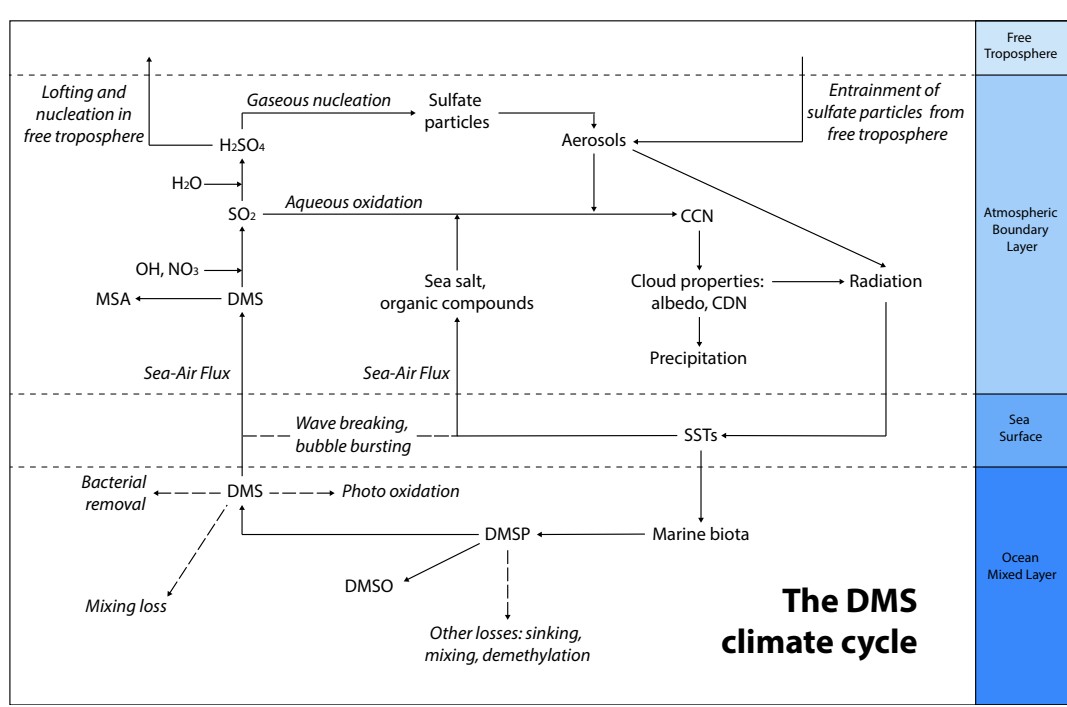

**Figure 1.** Schematic diagram showing the ocean-atmosphere sulfur life cycle and climate-relevant processes. Acronyms are defined as follows: sea surface temperatures (SSTs), methane sulfonic acid (MSA), dimethyl sulfoniopropionate (DMSP), dimethyl sulfoxide (DMSO), dimethyl sulfide (DMS), cloud condensation nuclei (CCN), cloud droplet number (CDN)



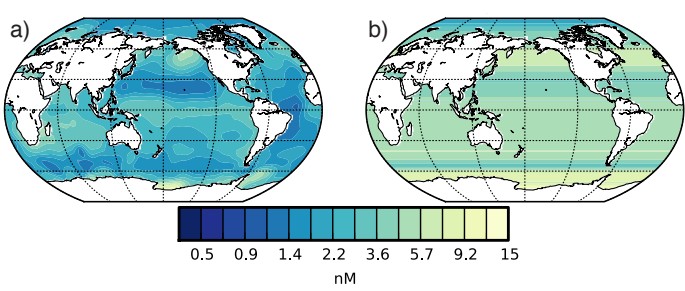

**Figure 2.** The annual mean concentrations of a) the Lana et al. (2011) $DMS_w$ climatology, b) the $DMS_w$ field of the second experimental run: zonal maximum $DMS_w$




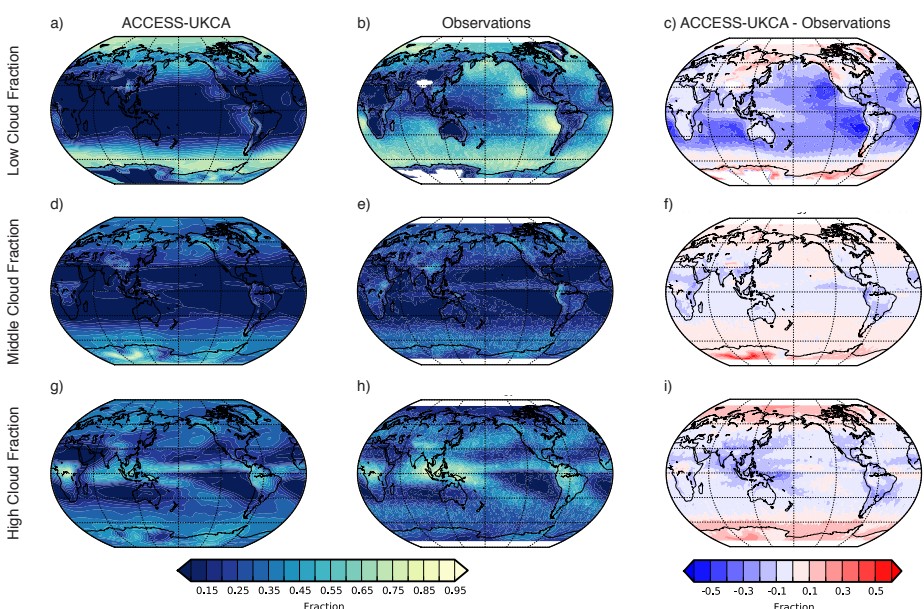

**Figure 3.** The 2006-2009 annual mean of the ACCESS-UKCA CTL (first column) compared to the CALIPSO-GOCCP (Chepfer et al., 2010) climatology (second column), with the absolute differences between the two shown in the third column. The top row shows the low cloud fraction, the middle row shows the middle cloud fraction and the bottom row the high cloud fraction



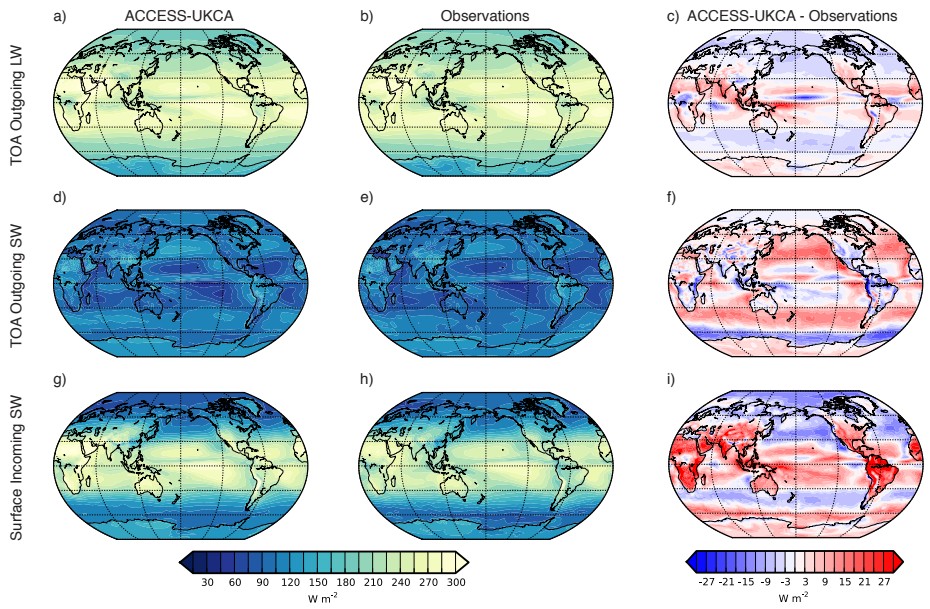

**Figure 4.** As for Fig. 3 but for annual means for 2000-2009, where the top row shows the TOA outgoing LW radiation and the second row: TOA outgoing SW radiation and the third row: surface incoming SW radiation. The observations are from the CERES-EBAF - TOA and Surface Ed. 4.0 (Loeb et al., 2009; Kato et al., 2013); all units are in $\mathrm{W\ m^{-2}}$





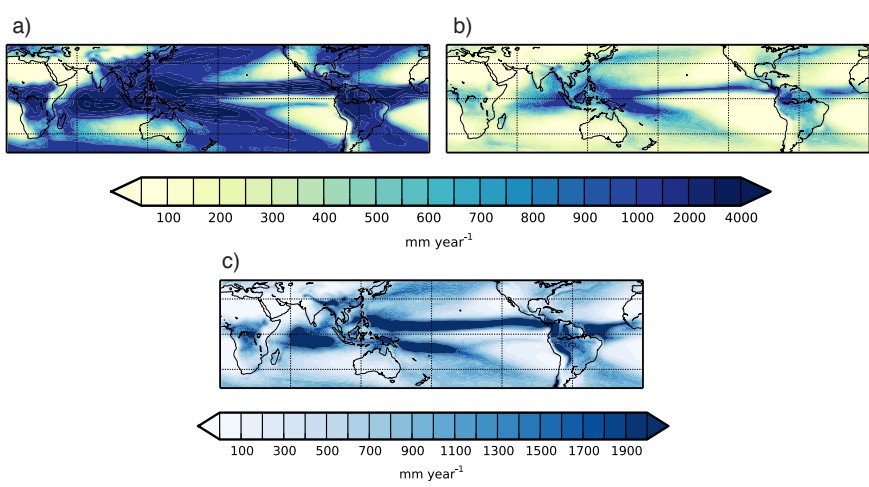

**Figure 5.** The mean (2000-2009) annual total precipitation of a) the satellite climatology from TRMM (Huffman et al., 2007)), b) the ACCESS-UKCA climatology and c) the difference between the TRMM and ACCESS-UKCA climatologies. Units are in $\mathrm{mm\ year}^{-1}$





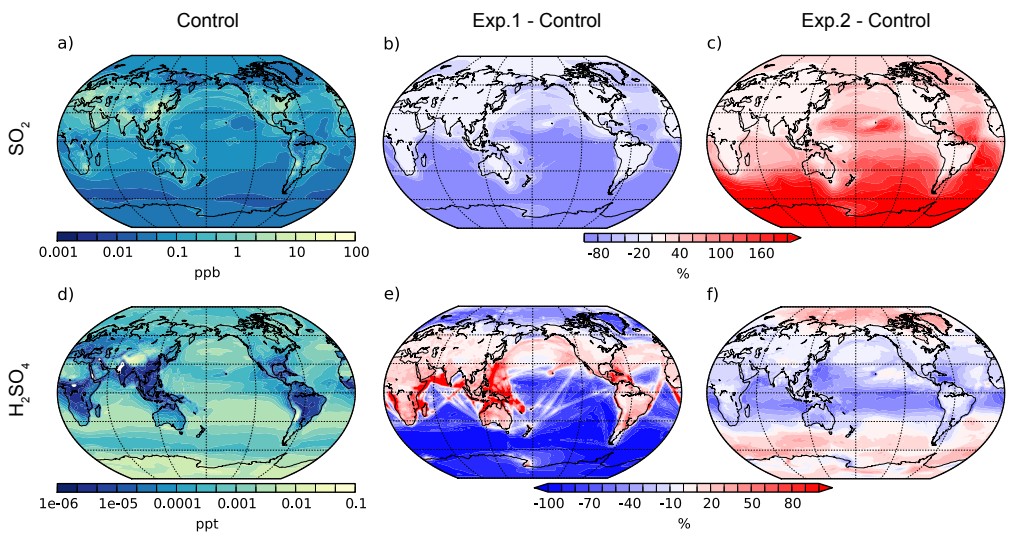

**Figure 6.** Mean (2000-2009) values for the Ctl (first column), the difference between Exp.1 (zero $DMS_w$) and the Ctl (second column) and the difference between Exp.2 (zonally enhanced $DMS_w$) and the Ctl (third column). The first row shows the volume mixing ratio of $SO_2$ in ppb; the second row the volume mixing ratio of $H_2SO_4$ in ppt



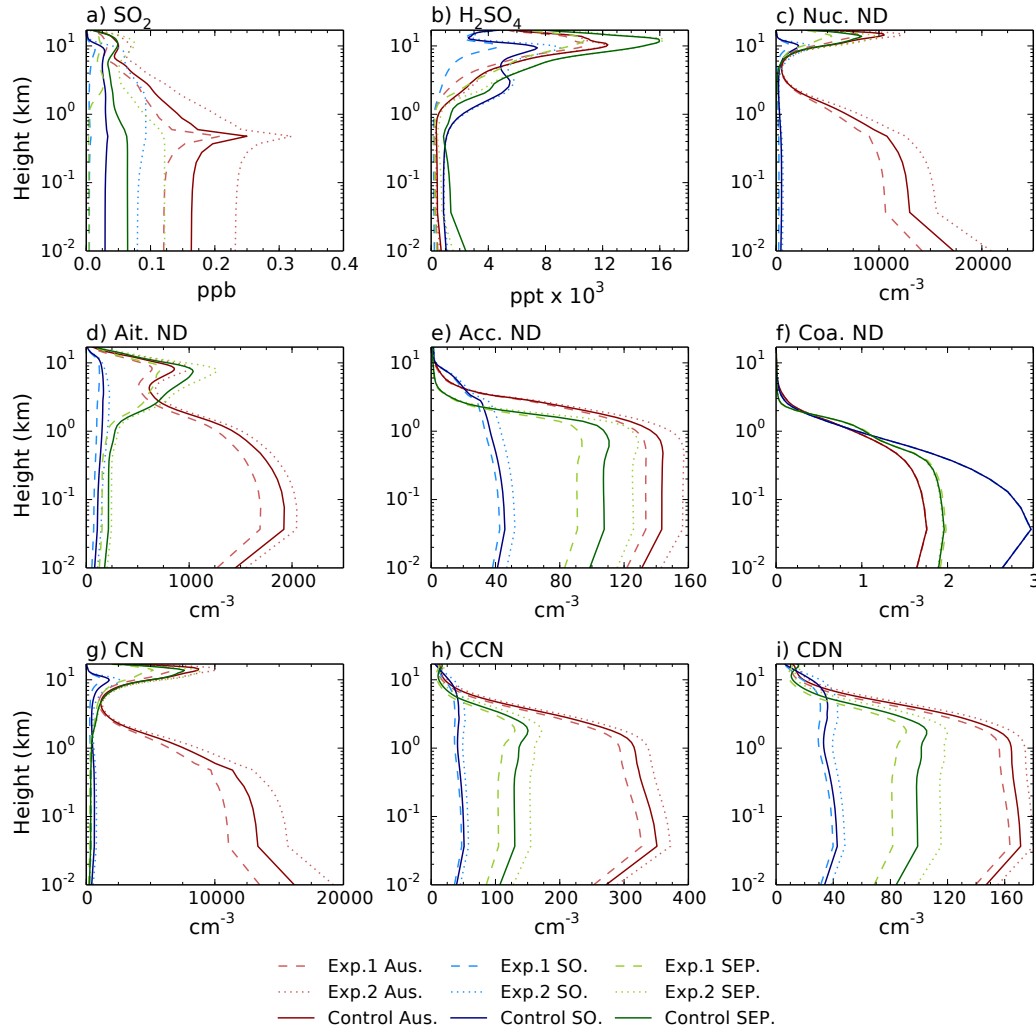

**Figure 7.** The vertical profiles of a) $SO_2$; b) $H_2SO_4$; c) nucleation mode number density; d) Aitken mode number density; e) accumulation mode number density; f) coarse mode number density; g) $N_3$ nuclei number; h) cloud condensation nuclei number; and i) cloud droplet number. The solid lines represent the Ctl; dashed lines shows Exp.1; and dotted lines show Exp.2. Blue lines show the SO (SO) mean, red the Australian (Aus) region mean and green the South Eastern Pacific (SEP). All units are $cm^{-3}$, apart from a): ppb and b); ppt x $10^{-3}$


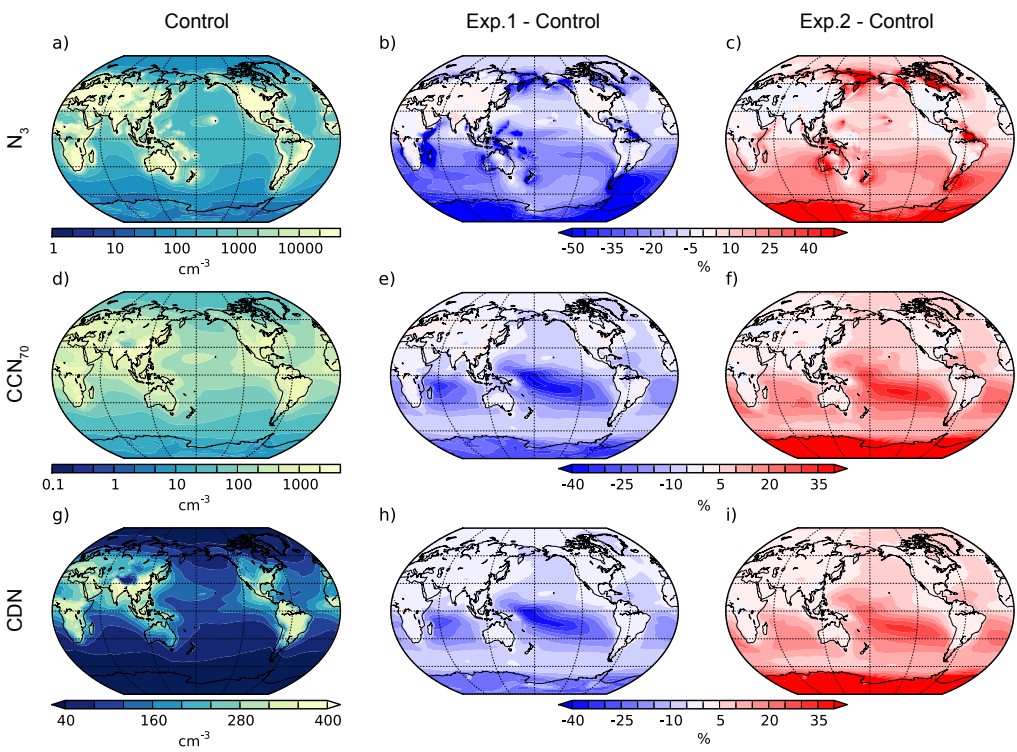

**Figure 8.** As for Figure 6, but where the first row shows the number concentration of $N_3$ (condensation nuclei); the second the number of cloud condensation nuclei greater than $70\,\mathrm{nm}$ dry diameter; and the third the cloud droplet number concentration. All units are in $\mathrm{cm}^{-3}$





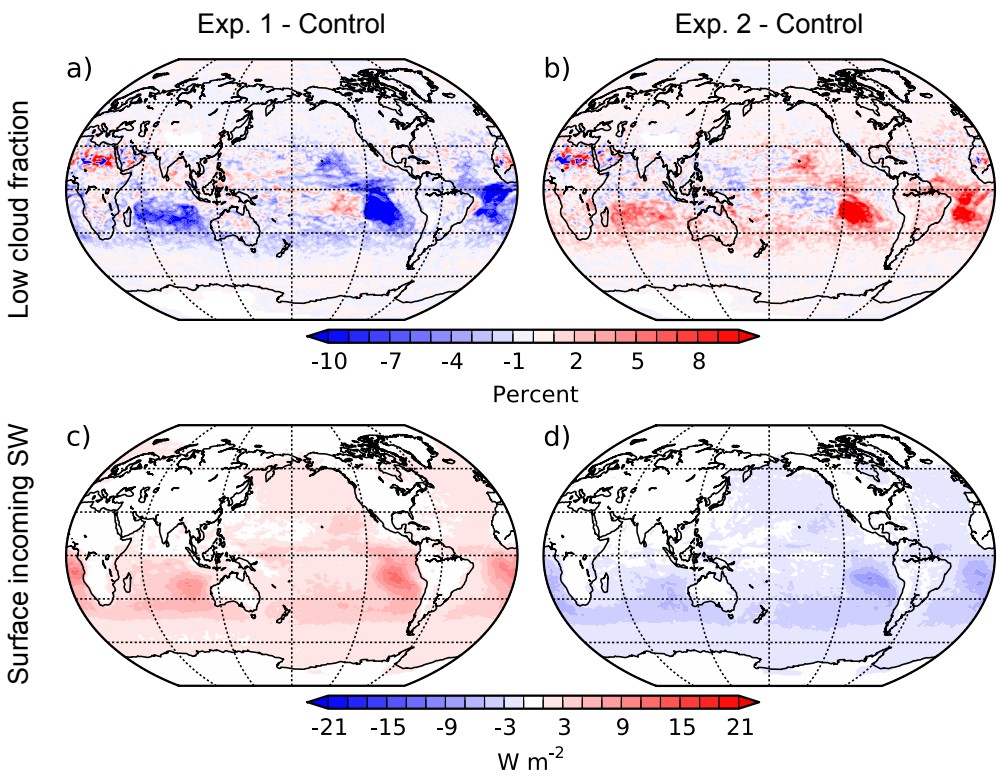

**Figure 9.** Comparisons in of the low cloud fraction (as a percentage) (a-b) and incoming SW radiation at surface (W m$^{-2}$) c-d) over the 2000-2009 period for Exp.1 (first column) and Exp.2 (second column) minus the Ctl. The absolute values for the Ctl of these fields can be seen in Fig. 3 and 4





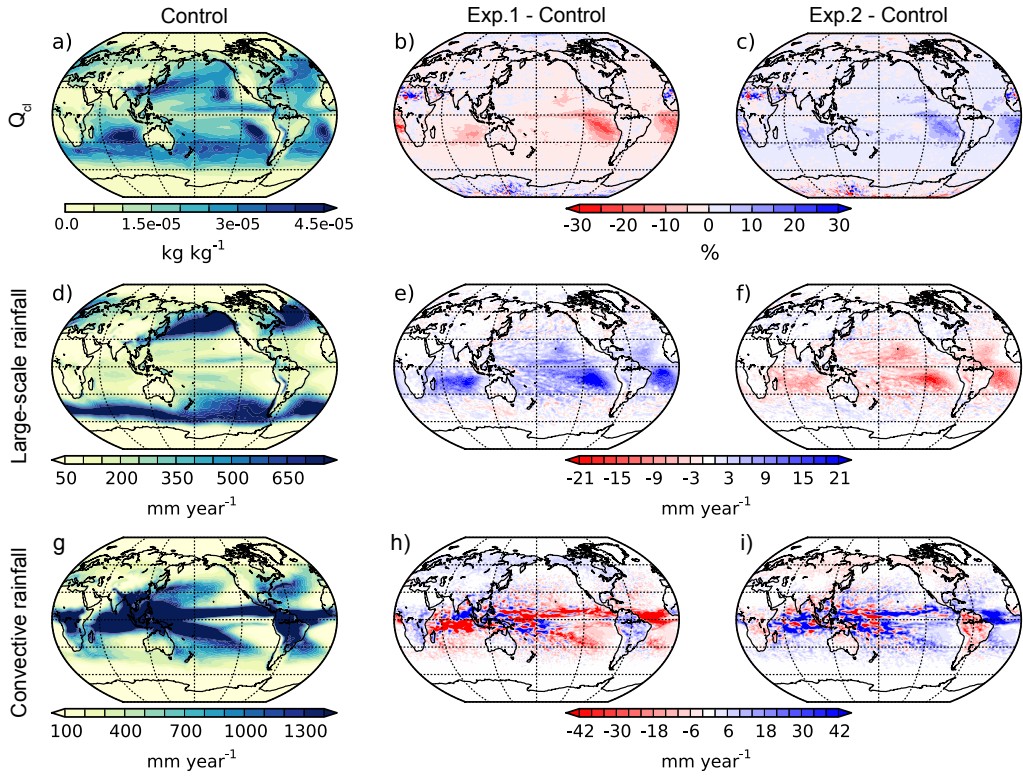

**Figure 10.** Comparisons of the (a-c) total liquid water at 1700m height ($Q_{cl}$), (d-f) large-scale rainfall, (g-i) convective rainfall over the 2000-2009 period. The Ctl absolute values are shown in the first column, and respectively Exp.1 and Exp.2 minus the Ctl in the second and third columns. Units are in $\mathrm{kg\,kg^{-1}}$ and $\mathrm{mm\,year^{-1}}$





**Table 1.** Summary of the three global simulations presented in this study, the $DMS_w$ climatology used and the annual mean (2000-2009) total global flux$_{DMS}$

| Simulation | DMS Climatology | Flux$_{DMS}$ (Tg year$^{-1}$ of sulfur) |
|---|---|---|
| Ctl | Lana et al. (2011) | 17.41 |
| Exp.1 | Zero marine DMS | 0.72 |
| Exp.2 | Zonal maximum DMS from Lana et al. (2011) | 37.05 |



**Table 2.** Global and hemispheric means of the CCN sensitivity to the flux$_{\mathrm{DMS}}$ (as defined by Woodhouse et al. (2010, 2013)) in both absolute ($\mathrm{cm}^{-3}/\mathrm{mg\,m^2\,day^{-1}}$) and relative terms, for Exp.1 and Exp.2.

| Region | Exp.1 Absolute | Exp.2 Absolute | Exp.1 Relative | Exp.2 Relative |
|--------|----------------|----------------|----------------|----------------|
| Global | 16.9 | 12.4 | 0.048 | 0.036 |
| SH | 15.8 | 11.2 | 0.090 | 0.063 |
| NH | 18.6 | 14.5 | 0.029 | 0.023 |



**Table 3.** Summary of the global mean (2000-2009) radiation fields: absolute Ctl values for the TOA shortwave (SW) and longwave (LW) outgoing and Q*; and the differences in these quantities resulting from Exp.1 and Exp.2 (from the Ctl) as well as the FAIR temperature response

| Simulation | TOA SW ↑ (W m$^{-2}$) | TOA LW ↑ (W m$^{-2}$) | Q* (W m$^{-2}$) | FAIR response (°K) |
|---|---|---|---|---|
| Ctl absolute values | 101.79 | 241.04 | -1.35 | NA |
| Exp.1 - Ctl | -1.82 | 0.13 | 1.69 | 0.45 |
| Exp.2 - Ctl | 1.57 | -0.12 | -1.45 | -0.38 |





**Table 4.** The estimated temperature response to perturbations in the $\text{flux}_{\text{DMS}}$ (K per Tg year$^{-1}$ of sulfur) for the current study's experiments (Exp.1 and Exp.2) and those found in the literature

| Experiment | K per Tg year$^{-1}$ of sulfur |
|---|---|
| Exp.1 | 0.027 |
| Exp.2 | 0.019 |
| Schwinger et al. (2017) | 0.041 |
| Six et al. (2013) - low pH impact scenario | 0.03 - 0.060 |
| Six et al. (2013) - medium pH impact scenario | 0.046 - 0.096 |
| Six et al. (2013) - high pH impact scenario | 0.051 - 0.11 |
| Grandey and Wang (2015) | 0.029 |