# Peer review of "Cloud, precipitation and radiation responses to large perturbations in global dimethyl sulfide"

_Atmospheric Chemistry and Physics, 2017_

## Referee Comment (RC1) · Anonymous Referee #1 · 6 Mar 2018

In this paper the authors presented the results from model simulations with the aim of quantifying the effects of changing DMS natural emissions on cloud, precipitation and radiation. They compared three global chemistry-climate coupled model simulations, a control run with the most recent climatology of DMS water concentrations, a sensitivity by removing all marine DMS, and a second sensitivity by imposing the monthly zonal maximum DMS water concentrations at each latitude. Despite this study does not present novel concepts, as similar experiments were recently performed by other studies, this paper address relevant scientific questions, which are not yet fully answered. The presentation and the analysis of the results is generally well written and clear, but it can be improved to clarify some of the methods and the analysis. I would suggest acceptance of the paper after taking into consideration the following comments.

[Figure]

General comments:

- In the abstract not all results are summarized, like the comparison of the control simulation with observations, the second experiment with increased DMS emissions, and the estimate of temperature response to changes in DMS TOA radiative effect. The final sentence of the abstract is too general, the authors should try to briefly explain what kind of model study are further needed and why.

- The explanation given at the end of section 2.1.1 about nudging is not very clear. I understand that doing nudged simulation is better to compare the control model simulation with observation. On the other hand the disadvantage is to not have the full impact on meteorology when comparing the control simulation with the two sensitivity experiments. As far as I understand the model simulation includes both direct and indirect effect of aerosols on radiation, so it is not very clear the sentence at page 4, lines 15-18. Maybe also the paragraph at page 14, lines 3-5 should be included in section 2.1.1.

- The quality of the figures is not always satisfying. In particular the figures which include multiple maps are too small and it is difficult to visualize the fields. Also sometime the colors does not help the data visualization. In particular I would recommend to improve Figures 2,3,4,5,6,8 and 10.

- The section 5 is too long and somehow difficult to read. I would recommend to split in two or to shorten it by removing some of the details which are repeated from the previous sections. I would try to explain better the last part of the discussion, providing more details on what kind of experiments are needed to better understand the role of DMS future, considering the ocean acidifcation, as mentioned in the last paragraph.

Minor comments:

Page 3, line 1: Six et al. (2013)

Page 3, Line 21: the control simulation is not explained, only the two DMS perturbation

simulations are described.

Page 4, Line 1: In this study, . . .

Page 4, Line 10: did you forget CO2 from the list of GHG gases?

Page 6, Line 1: 50-440 hPa

Page 6, Line 17-19: WHy these three regions were chosen? A short motivation should be added. I would include the boundaries of these regions in one of the figures.

Page 6, Line 18: Pacific

Page 6, Line 29: for the first time the CTL name is used, should be introduced before

Page 6, Line31: "without the need for an . . .", I would remove or rephrase as an ensemble experiment of free-running climate simulations is needed to better quantify the impact of DMS forcing on the temperature.

Page 7, Line 9: fraction larger then 0.5 instead of >

PAge 8, Line 18: outgoing TOA, LW or SW?

Page 8, Line 30: (Fig 5c)? positive bias if is the difference between the model and observations, not clear from figure caption.

Page 8, Line 30: over regions to the north and south if the equator, .. but only in the tropics is over 2000 mm/yr

Page 9, Line 4: the figure with % differences is not shown, but the values of the largest % differences could be inserted in the text.

Page 9, Line 17: The largest absolute differences are in the tropics and mid-latitudes over the Oceans.

Page 10, Line 5: Fig6 g-h is not correct

Page 10, Line 21: the largest absolute differences are in clean terrestrial regions.

[Figure]

Which regions? Not easy to visualize in the figures. Too small.

Page 11, Line 11: the results presented here suggest a lower CCN . . .

Page 11,Line 18: would title the section "Clouds and precipitation response"

Page 11, Line 21-23: this explanation about nudging should be explained also before when describing the experiments.

Page 11, Line 32: (see section 3)

Page 12, Line 4: remove ) after fig 10a-b

Page 13, Section 4.3: Is it possible to put the uncertainties of these estimates of temperature changes per Tg of S emitted?

Page 14, Line 33: put ref of thomas and mahajan in parenthesis

Page 16, Line 3: can you better explain the role of coral-reef derived DMS? Why it is important?

Page 16, Line 8: our results imply that a 25% decrease in . . . would result in an increase of 0.1C. Is it possible to put the uncertainty on this estimate?

Caption of figure 3: third column is not absolute differences, as negative numbers are shown.

Caption Figure 5: c) not clear if difference model-obs or contrary while reading the description of the figure in the manuscript.

Caption Figure 7: blue lines show the SO (Southern Ocean) mean, red the australian (aus). The short name aus is used only here and not in the manuscript.
* * *

---

## Referee Comment (RC2) · Anonymous Referee #2 · 9 Mar 2018

The manuscript reports on the influence of natural aerosol emissions on climate systems. Specifically, it presents research on the relationship between biologically-derived dimethyl sulfoxide (DMS) and climatic parameters (e.g., cloud cover, radiation and precipitation). In addition, the manuscript provides an evaluation of the ACCESS UKCA model for examining the role of DMS in the Earth's climate, and conducted large-scale sensitivity test using ACCESS UKCA modeling system in investigation of response of climate on prescribed changes in surface DMS concentrations. The relationship between DMS and climate processes is strongly non-linear and this relationship is poorly understood due to lack of global DMS distribution data. Therefore, the research reported in the manuscript is timely and relevant.

The research presented in the manuscript had the following objectives: (1) to evalu-

ate the application of the ACCESS UKCA model for examining the role of DMS in the Earth's climate and (2) to conduct large-scale sensitivity test using the ACCESS UKCA modeling system in investigation of climate response to prescribed changes in surface water DMS. The manuscript is well written and provides new and interesting analyses that are useful for the scientific community. It is also well structured and the methodology is consistent with the objectives of the study. I recommend it for publication following suggested revisions as described below.

A number of published studies have investigated the response of the climate system to changes in DMS flux, using modeling approaches with similar aims of understanding the effect of DMS on the global system. Therefore, the manuscript will be greatly improved by clearly stating specifically what the study contributes in addition what is already found in the literature.

In accordance to above, the Introduction and Discussion should provide a balanced and useful discussion of the results in the context of previously published work. In its present form the manuscript fails to cite a number of relevant papers which are noted below in the point-by-point comments.

Also, the problem statement is not well defined and should be improved by answering what this study makes it novel from previous studies. Only using a different GCM model (i.e. here ACCESS UKCA) does not necessarily bring originality to the study. So there should be more mentioned about what advantages the considered model bring compared to other models in terms of physics and chemistry schemes. In sensitivity tests only two extreme cases are considered (i.e., zero and zonal maximum). As already mentioned, the response of the climate to DMS is highly non-linear, such that it would be reasonable to have a better understanding of how climate responds to smaller change in DMS.

Point by point comments on the manuscript follow:

Introduction: The introduction provides relevant background regarding the role of

aerosols in the global radiation budget. It would benefit from more background information regarding the uncertainty in DMS concentrations in the ocean. See for example: Belviso et al. (2004), Tesdal et al. (2016a).

(page 2 line 9): Additional support for the use of DMS fields in climate model are given in Belviso et al. (2004) and Tesdal et al. (2016a), and it is recommended to include these references. Hopkins et al. (2016) is not an appropriate reference for the uncertainty around observed DMS concentrations. Royer et al. (2015) is suggested.

(page 2 line 10) Charlson et al (1987) is not an appropriate reference for the role of DMS in climate systems being subject to debate, as this is the paper that introduced the hypothesis. The debate came later. A more appropriate article that also serves as an review of the CLAW hypothesis is Ayers and Cainey (2007).

(page 2 line 16) Add "s" to "contribute"

(page 2 line 23) Add "in certain regions" following "...local DMS concentrations"

(page 2 line 29) Add "in order" between "... a one year simulation" and "to quantify its importance..."

(page 2 line 32) Replace "for example" with "e.g."

(page 2 line 34) Is ocean acidification meant to be an example of "anthropogenic climate change"? Please clarify.

(page 3 line 3 and throughout manuscript) Both Celsius and Kelvin temperature units are used throughout. Please use one or the other (preferably Celsius) for consistency.

(page 3 line 10) Replace "low-mid level" with "low- and mid-level"

(page 3 line 26) Add "analyzes" after "Section 4"

(page 4 line 23) Replace "of" with "detailed in"

(page 4 line 28) " A full description of the scheme can be found in Mann et al..." Here

it would be appropriate to mention the study by Bellouin et al. (2013), who compared GLOMAP-mode and CLASSIC and determine that GLOMAP-mode is more sophisticated and accurate.

(page 5 line 1) "Significant sampling biases...Northern Hemisphere (Lana et al 2011)" This sentence is not entirely clear. Is the point that the data is biased towards spring-summer and towards Northern Hemisphere?

(page 5 line 2) Replace "spring-summer" with "spring through summer"

(page 5 line 5) "The Liss and Merlivat (1986) parameterization..." There should be justification for why this parameterization is used. Several different parameterizations of the piston velocity in terms of wind speed have been used in modelling studies (e.g., Liss and Merlivat, 1986; Wanninkhof, 1992; Nightingale et al., 2000), leading to substantially different flux fields for a given concentration field (Tesdal et al., 2016a).

(page 5 line 16-17) Following Eq. 4, Replace "Where" with "Here," . Insert "is" before "determined following the method of Saltzman.."

(page 5 line 20) "seawater" should not be hyphenated (Section 2.2 Model Evaluation) The evaluation should include comparison to observation of atmospheric concentration of DMS and other sulfur species, for example as described in Tesdal et al. (2016b).

(page 6 line 1) add "between" between "medium" and "440-680 hPa" and between "low" and "680-1000 hPa"

(page 6 line 17) Add period at end of sentence that begins "The Australian region...."

(page 6 line 24) Replace "that" with "the one"

(page 6 line 30) "By providing this radiative effect..." The text appears to imply that FAIR is a tool that provides estimates of climate response, given a simple value of radiative effect. Thus, by feeding estimates of radiative effects to FAIR one can analyze the effect on temperature and other climate variables.The text would benefit from more

elaboration regarding the relationship between the radiative effect and the FAIR climate component.

(page 7 line 12) Add comma after "e.g."

(page 8 line 12) "...instead reflecting more sunlight thus enhancing the albedo..." This clause is confusing. What is it that is reflecting more sunlight?

(page 8 line 20) Replace "...low clouds allowing.." with "...low clouds, which allows..."

(page 8 line 24) "This estimation is slightly greater..." Greater than what?

(page 9 line 20) Replace "representing" with "which is due to" or "which represents" to avoid awkward verb-gerund construction ("...representing lofting...").

(page 9 line 23) Use appropriate subscript formatting for H2SO4.

(page 9 line 25) Replace "...in new particle formation, forming secondary sulfate aerosol..." with "...in formation of secondary sulfate aerosols,..."

(page 9 line 26) Correct formatting of H2SO4.

(page 10 line 5 and throughout manuscript) Replace "Whilest" " with "While"

(page 10 line 26) Replace "cloud condensation nucleii" with "CCN"

(page 10 line 26-33) "Fig. 8d-e show the number..." The paragraph does not clearly describe the figures; sentence structure is awkward. Consider revision as follows: Fig. 8d-e show the number concentration of CCN with dry diameters greater than 70 nm (CCN70) for the Ctl and the differences resulting from Exp.1. The largest absolute differences are in the tropics, which, similarly to the N3, have the highest concentration. Relatively, there is a global decrease of 5%, whilst decreases of 7% were found over the Australian region, decreases of 8% over the SO and decreases of 20% over the SEP. Differences in cloud droplet number (CDN) are shown in Fig. 8g-h. The relative differences in CDN (Exp.1-Ctl) show a similar spatial pattern to that of the CCN. Global

mean CDN decreases by 5%. A decrease of 5% is also found for the Australian region, whereas the SO shows an 8% decrease, and the SEP shows an 18% decrease. In both the CCN70 and CDN, the marine Southern Hemisphere mid-latitudes have the largest decreases of 14% (averaged between 5-35ËŽS) despite the SO having some of the larger decreases in SO2 and H2SO4.

(page 10 line 30) "Exp 1- Ctl" Is this explaining the ratio for relative difference? It is not given elsewhere when talking about relative difference.

(page 11 line 1) Replace "tropics mid-latitudes" with "mid-latitude tropics".

(page 11 line 13) "Similar CCN sensitivities are reported in the Woodhouse et al. (2010) study." State actual values reported in the reference.

(page 11 line 24) "...radiation scheme via aerosols and some (see Section 2.1.1)." Missing word/phrase following "some".

(page 11 line 28) "...slightly removed from the coastline." Not clear. From the Southern Hemisphere's coastline? Recommend revising the whole first sentence, ending with a period after "(Fig. 9a)" and then beginning a new sentence. This would help to clarify what difference is seen in the SH generally and would parallel the structure used for the rest of the paragraph.

(page 11 line 32) "...see Section 3 in comparison to other areas..." Confusing. The author is comparing all of Section 3 to the areas of low cloud formation?

(page 11 line 35) "...biases exacerbate this..." Clarify what "this" refers to.

(page 12 line 4) Complete parentheses for "shown in Fig. 10a-b"

(page 12 line 8) Capitalize "Southern Hemisphere"

(page 12 line 13) Add "the" before "Ctl"

(page 12 line 25) Delete the comma after "magnitude"

(page 12 line 28) Revise text to "...the SEP: increases of 42%, 172%, and 89% respectively."

(page 12 line 29) Insert "occurs" after "...a decrease of 37%"

(page 12 line 33) Insert "respectively" after "...increases by 6%, 4% and 5%"

(page 12 line 34) Insert "the" before "...SEP of 14%"

(page 13 line 2) Replace "showing" with "which show"

(page 13 line 2) Suggest "Incoming surface SW radiation" rather than "Surface incoming SW radiation..."

(page 13 line 6) Replace "analyzed under" with "which analyzed"

(page 13 line 8) Replace "south east" with "southeast"; Replace "these results presented" with "the results presented here"

(page 13 line 10) Insert "however" between "...the surface," and "the precipitation..."

(page 13 line 18) Delete "Although"; insert "and" between "...warming" and "we..."

(page 13 line 21) Delete comma after "...study"

(page 13 line 26) Replace "on" with "in"

(page 13 line 30) Delete duplicate "the"; insert "that" between "...this study is" and "the model..."; replace "underestimation of" with "underestimates"

(page 14 line 1) Replace "rather a multitude of theories" with "multiple theories have been proposed"

(page 14 line 7) "representation" should be plural

(page 14 line 13) Change "Figure 1" to "Fig. 1" for consistency with rest of manuscript

(page 14 line 14) Delete comma after SO2

(page 14 line 32) Break run-on sentence: Period after "importance". Next sentence: "Instead, the studies highlighted SO (Thomas et al., 2010; Mahajan et al., 2015)." Highlighted the SO in terms of what parameter(s)?

(page 14 line 35) Revise sentence as: "...slightly lower than the estimation of 2.03 Wm$-2$ by Thomas..."

(page 15 line 1) Replace "are" with "were"

(page 15 line 5) Revise reference format: (Albrecht, 1989)

(page 15 line 16) Revise placement of subordinate clause in the sentence beginning "The cause of the discrepancy..." as: "Without further information, it is difficult to speculate on the cause of the discrepancy..."

(page 15 line 18) "Suggest" should be plural; delete comma after constraints

(page 15 line 22) Replace "Whilst" with "Though"

(page 15 line 23) Replace the colon after "For example" with a comma; replace "an as of yet" with currently; replace "for" with "as a"

(page 15 line 31) Replace the semicolon after "Six et al (2013) with "and"

(page 16 line 3) "coral-reef-derived" (2 hyphens), or "DMS derived from coral reefs"

(page 16 line 5) Replace semicolon after "Hopkins et al. (2011)" with a comma

(page 16 line 8) Insert "those" between "than" and "found"

(page 16 line 12) Make example given in this sentence parenthetical: "...increases (e.g., via solar radiation management) may have a short term cooling effects, however, without..."

(page 16 line 13) Delete the "a" between "...may have" and "short term..."; insert comma after "...however"

[Figure]

(page 16 line 14) Delete "on" between "...impact" and "marine life..."

References

Ayers, G. P., & Cainey, J. M. (2007). The CLAW hypothesis: a review of the major developments. Environmental Chemistry, 4(6), 366–374. http://doi.org/10.1071/EN07080

Bellouin, N., Mann, G. W., Woodhouse, M. T., Johnson, C., Carslaw, K. S., & Dalvi, M. (2013). Impact of the modal aerosol scheme GLOMAP-mode on aerosol forcing in the Hadley Centre Global Environmental Model. Atmospheric Chemistry and Physics, 13(6), 3027–3044. http://doi.org/10.5194/acp-13-3027-2013

Belviso, S., Bopp, L., Moulin, C., Orr, J., Anderson, T., Aumont, O., et al. (2004). Comparison of global climatological maps of sea surface dimethyl sulfide. Global Biogeochemical Cycles, 18(3). http://doi.org/10.1029/2003GB002193

Nightingale, P., Malin, G., Law, C., Watson, A., Liss, P., Liddicoat, M., et al. (2000). In situ evaluation of air-sea gas exchange parameterizations using novel conservative and volatile tracers. Global Biogeochemical Cycles, 14(1), 373–387. http://doi.org/10.1029/1999GB900091

Royer, S.-J., Mahajan, A. S., Galí, M., Saltzman, E., & Simó, R. (2015). Small-scale variability patterns of DMS and phytoplankton in surface waters of the tropical and subtropical Atlantic, Indian, and Pacific Oceans. Geophysical Research Letters, 42(2), 475–483. http://doi.org/10.1002/2014GL062543

Tesdal, J.-E., Christian, J. R., Monahan, A. H., & Salzen, von, K. (2016a). Evaluation of diverse approaches for estimating sea-surface DMS concentration and air–sea exchange at global scale. Environmental Chemistry, 13(2), 390–412. http://doi.org/10.1071/EN14255

Tesdal, J.-E., Christian, J. R., Monahan, A. H., & Salzen, von, K. (2016b). Sensitivity of modelled sulfate aerosol and its radiative effect on climate to ocean DMS concentration and air–sea flux. Atmospheric Chemistry and Physics, 16(17), 10847–10864.

http://doi.org/10.5194/acp-16-10847-2016

Wanninkhof, R. (1992). Relationship Between Wind Speed and Gas Exchange Over the Ocean. Journal of Geophysical Research, 97(C5), 7373–7382. http://doi.org/10.1029/92JC00188

---

## Author Comment (AC1) · 23 Apr 2018

**Response to Reviewer 1**

We would like to thank Reviewer 1 for their comments on our manuscript. We agree that while similar simulations have been conducted, our focus has been on providing a complete end-to-end evaluation of the role of DMS in the climate system, from changes in the chemistry through to effects on clouds, precipitation and radiation. Furthermore, we have highlighted the problems remaining when trying to model and understand this complex system. We have made these discussion points clearer in both the introduction and conclusion, which has strengthened the arguments presented in this work. Furthermore, we have clarified our methods, which hopefully leaves no remaining am-

biguity. We have addressed the specific comments below, and we hope you find our revisions satisfactory.

**General Comments:**

*In the abstract not all results are summarized, like the comparison of the control simulation with observations, the second experiment with increased DMS emissions, and the estimate of temperature response to changes in DMS TOA radiative effect. The final sentence of the abstract is too general, the authors should try to briefly explain what kind of model study are further needed and why.*
We have revised the abstract to include the model evaluation, results of both experiments and the temperature sensitivity to DMS flux. We have also made the abstract more specific.

*The explanation given at the end of section 2.1.1 about nudging is not very clear. I understand that doing nudged simulation is better to compare the control model simulation with observation. On the other hand the disadvantage is to not have the full impact on meteorology when comparing the control simulation with the two sensitivity experiments. As far as I understand the model simulation includes both direct and indirect effect of aerosols on radiation, so it is not very clear the sentence at page 4, lines 15-18. Maybe also the paragraph at page 14, lines 3-5 should be included in section 2.1.1.*
We have revised our explanation and justification of using nudging in our simulations in the methods section as suggested. The reviewer is correct that the model can simulate direct and indirect aerosol effects. Our use of the word 'direct' in this section may have been misleading and we have rectified this. We have also revised the text on the use of nudging in the discussion section.

*The quality of the figures is not always satisfying. In particular the figures which include multiple maps are too small and it is difficult to visualize the fields. Also sometime the colors does not help the data visualization. In particular I would recommend to improve*

*Figures 2,3,4,5,6,8 and 10.*
We have adjusted the colour scales by removing the darkest colours to increase visibility of the figures. We have also removed white space where we can and enlarged the figures. We hope these changes make our figures easier to interpret. We will also ensure the figures are readable after the typesetting process.

*The section 5 is too long and somehow difficult to read. I would recommend to split in two or to shorten it by removing some of the details which are repeated from the previous sections. I would try to explain better the last part of the discussion, providing more details on what kind of experiments are needed to better understand the role of DMS future, considering the ocean acidifcation, as mentioned in the last paragraph.*
We have significantly shortened this section, cutting out repetitive paragraphs summarising results previously discussed as suggested by the reviewer. We strengthened our discussion involving not only the questions we are addressing but also pointing towards what steps need to be taken for future studies. We are hesitant to include more information about the impacts of ocean acidification as it is beyond the scope of this study (and admittedly, not our area of expertise). However, we do wish to highlight potential implications of ocean acidification on climate that may not have previously been considered.

**Minor Comments:**

*Page 3, line 1: Six et al. (2013)*
Typo has been amended

*Page 3, Line 21: the control simulation is not explained, only the two DMS perturbation C2 simulations are described*
We have included a description of the control simulation

*Page 4, Line 1: In this study, . . .*
We are unsure what the reviewer meant here. We do not feel starting the paragraph with 'in this study' is appropriate.

*Page 4, Line 10: did you forget CO2 from the list of GHG gases?*
CO2 has now been explicitly mentioned, it is prescribed to the model as global mean concentration.

*Page 6, Line 1: 50-440 hPa*
Typo has been amended

*Page 6, Line 17-19: WHy these three regions were chosen? A short motivation should be added. I would include the boundaries of these regions in one of the figures.*
We have included motivation as to why these regions have been chosen and included the boundaries in Figure 2.

*Page 6, Line 18: Pacific*
Typo has been amended

*Page 6, Line 29: for the first time the CTL name is used, should be introduced before*
The Ctl name has been introduced a few paragraphs above.

*Page 6, Line31: "without the need for an . . .", I would remove or rephrase as an ensemble experiment of free-running climate simulations is needed to better quantify the impact of DMS forcing on the temperature.*
We have rephrased this sentence as suggested

*Page 7, Line 9: fraction larger then 0.5 instead of >*
Typo has been amended

*Page 8, Line 18: outgoing TOA, LW or SW?*
We have clarified this (SW)

*Page 8, Line 30: (Fig 5c)?  positive bias if is the difference between the model and observations, not clear from figure caption.*
We have revised the figure caption for clarity

*Page 8, Line 30: over regions to the north and south if the equator, .. but only in the*

*tropics is over 2000 mm/yr*
We have revised this sentence for clarity

*Page 9, Line 4: the figure with % differences is not shown, but the values of the largest % differences could be inserted in the text.*
We have included the % differences in the text as suggested

*Page 9, Line 17: The largest absolute differences are in the tropics and mid-latitudes over the Oceans.*
Sentence amended as suggested

*Page 10, Line 5: Fig6 g-h is not correct*
Typo has been amended

*Page 10, Line 21: the largest absolute differences are in clean terrestrial regions. C3 Which regions? Not easy to visualize in the figures. Too small.*
We have removed reference to these differences due to the difficulty of visualisation.

*Page 11, Line 11: the results presented here suggest a lower CCN . . .*
Sentence amended as suggested

Page 11,Line 18: would title the section "Clouds and precipitation response" Title amended as suggested

*Page 11, Line 21-23: this explanation about nudging should be explained also before when describing the experiments.*
We have revised our explanation of nudging in the methods section.

*Page 11, Line 32: (see section 3) Page 12, Line 4: remove ) after fig 10a-b*
Typo has been amended

*Page 13, Section 4.3: Is it possible to put the uncertainties of these estimates of temperature changes per Tg of S emitted?*
We have performed a moving block bootstrap to determine the uncertainty of the mean

changes in radiation, via the 10th and 90th percentile confidence intervals. This can then be translated into a change in temperature by the FAIR model and applied to the flux sensitivity calculations. Further description of how this was performed is now included in the methods and Tables 3 and 4 have been updated to include these uncertainties.

*Page 14, Line 33: put ref of thomas and mahajan in parenthesis*
Typo has been amended

*Page 16, Line 3: can you better explain the role of coral-reef derived DMS? Why it is important?*
We have included a sentence describing why coral reef derived DMS is important (the fact that it is unaccounted for in current global modelling). The role of coral reef DMS is not yet quantified and represents an opportunity for further research. Note that this section has been significantly revised as suggested in the general comments.

*Page 16, Line 8: our results imply that a 25% decrease in . . . would result in an increase of 0.1C. Is it possible to put the uncertainty on this estimate?*
We have now included confidence intervals for our temperature/flux sensitivity calculations, and the estimates of potential temperature change with decreasing DMS flux.

*Caption of figure 3: third column is not absolute differences, as negative numbers are shown.*
Caption has been amended

*Caption Figure 5: c) not clear if difference model-obs or contrary while reading the description of the figure in the manuscript.*
We have revised the caption for clarity

*Caption Figure 7: blue lines show the SO (Southern Ocean) mean, red the australian (aus). The short name aus is used only here and not in the manuscript.*
We have only shortened Australia to Aus. for clarity in this figure.

---

## Author Comment (AC2) · 23 Apr 2018

**Response to Reviewer 2**

We would like to thank Reviewer 2 for their extensive comments. We acknowledge that there are a number of recent studies examining the role of DMS in the climate system. Our study provides a thorough end-to-end analysis of all aspects of the chemical, aerosol and meteorological impacts of large DMS perturbations in the current climate. To our knowledge, previous studies have not considered the entire system, and typically only considered short time periods (1yr) or future projections.

We have taken the Reviewer's advice and have strengthened not only our problem statements in the introduction but also the points mentioned above in the introduction

and conclusion. We agree that just using a different model does not provide sufficient novelty to a study. We have now included some more information about why ACCESS-UKCA, but more particularly GLOMAP-mode, is a desirable tool, and further elucidated the novel aspects of the study in the manuscript

We also agree that the experiments performed here are extreme cases. However, there are significant insight into the DMS-climate system gained from these experiments that will serve to inform more targeted, realistic scenario experiments. Specifically, our large scale perturbations allow quantification of the overall contribution of DMS to the climate system.

We would also like to thank Reviewer 2 for bringing to our attention the two Tesdal et al. 2016 papers. They have provided a useful reference for discussion around DMS climatology and flux uncertainties.

**Specific comments:**

*The introduction provides relevant background regarding the role of aerosols in the global radiation budget. It would benefit from more background information regarding the uncertainty in DMS concentrations in the ocean. See for example: Belviso et al. (2004), Tesdal et al. (2016a).*
We have included further background information regarding the uncertainties surrounding the DMS climatology and flux as suggested by the Reviewer. We thank the Reviewer for their recommendations.

*(page 2 line 9): Additional support for the use of DMS fields in climate model are given in Belviso et al. (2004) and Tesdal et al. (2016a), and it is recommended to include these references. Hopkins et al. (2016) is not an appropriate reference for the uncertainty around observed DMS concentrations. Royer et al. (2015) is suggested.*
We have revised this section and included more appropriate references.

*(page 2 line 10) Charlson et al (1987) is not an appropriate reference for the role of*

*DMS in climate systems being subject to debate, as this is the paper that introduced the hypothesis. The debate came later. A more appropriate article that also serves as an review of the CLAW hypothesis is Ayers and Cainey (2007).*
We have included the recommended citation as the Reviewer has suggested

*(page 2 line 16) Add "s" to "contribute"*
Typo amended

*(page 2 line 23) Add "in certain regions" following "...local DMS concentrations" (page 2 line 29) Add "in order" between "... a one year simulation" and "to quantify its importance. . ."*
Sentence amended as suggested

*(page 2 line 32) Replace "for example" with "e.g."*
Sentence amended as suggested

*(page 2 line 34) Is ocean acidification meant to be an example of "anthropogenic climate change"? Please clarify.*
This sentence has been clarified to reflect that ocean acidification is caused by anthropogenic emission of $CO_2$.

*(page 3 line 3 and throughout manuscript) Both Celsius and Kelvin temperature units are used throughout. Please use one or the other (preferably Celsius) for consistency.*
We have made all temperature units consistent throughout the text

*(page 3 line 10) Replace "low-mid level" with "low- and mid-level" (page 3 line 26) Add "analyzes" after "Section 4"*
Sentence amended as suggested

*(page 4 line 23) Replace "of" with "detailed in"*
Sentence amended as suggested

*(page 4 line 28) " A full description of the scheme can be found in Mann et al. . ." Here it would be appropriate to mention the study by Bellouin et al. (2013), who*

*compared GLOMAP-mode and CLASSIC and determine that GLOMAP-mode is more sophisticated and accurate.*

We have made reference to the Bellouin et al. (2013) paper, as well as the Mann et al (2012) paper. We have retained the Mann et al. (2010) citation as this is the model description paper.

*(page 5 line 1) "Significant sampling biases...Northern Hemisphere (Lana et al 2011)" This sentence is not entirely clear. Is the point that the data is biased towards spring-summer and towards Northern Hemisphere?*

Yes, we have clarified this sentence to reflect this.

*(page 5 line 2) Replace "spring-summer" with "spring through summer"*

Sentence amended as suggested

*(page 5 line 5) "The Liss and Merlivat (1986) parameterization. . ." There should be justification for why this parameterization is used. Several different parameterizations of the piston velocity in terms of wind speed have been used in modelling studies (e.g., Liss and Merlivat, 1986; Wanninkhof, 1992; Nightingale et al., 2000), leading to substantially different flux fields for a given concentration field (Tesdal et al., 2016a).*

We have made reference to the alternative flux parameterisations as well as the modelling studies that detail their uncertainties in the introduction. Of the parameterisations available within ACCESS-UKCA, we chose the Liss Merlivat (1986) method because it agrees more closely with the latest generation of DMS flux parameterisations, e.g. Vlahos Monahan (2009), Bell et al. (2017), which were specifically derived for DMS, unlike the above parameterisations.

*(page 5 line 16-17) Following Eq. 4, Replace "Where" with "Here," . Insert "is" before "determined following the method of Saltzman.."*

Sentence amended as suggested

*(page 5 line 20) "seawater" should not be*

hyphenated Typo amended

*(Section 2.2 Model Evaluation) The evaluation should include comparison to observation of atmospheric concentration of DMS and other sulfur species, for example as described in Tesdal et al. (2016b).*

We appreciate that an evaluation of atmospheric DMS and other sulfur species is desirable, as done in the Tesdal et al. (2016b) study. However, we have limited this evaluation to meteorological data sets that are globally available from satellite products. We are confident in the use of this model for DMS studies due GLOMAP-mode being extensively evaluated against observations in many previous studies (Mann et al. 2010, 2012, Woodhouse et al 2010, 2013, 2015).

*(page 6 line 1) add "between" between "medium" and "440-680 hPa" and between "low" and "680-1000 hPa"*

Sentence amended as suggested

*(page 6 line 17) Add period at end of sentence that begins "The Australian region...."*
*(page 6 line 24) Replace "that" with "the one"*

Sentence amended as suggested

*(page 6 line 30) "By providing this radiative effect. . ." The text appears to imply that FAIR is a tool that provides estimates of climate response, given a simple value of radiative effect. Thus, by feeding estimates of radiative effects to FAIR one can analyze the effect on temperature and other climate variables.The text would benefit from more elaboration regarding the relationship between the radiative effect and the FAIR climate component.*

We have included some more information on how the FAIR emulator links radiative forcing changes to temperature and we have included some additional references for readers to pursue.

*(page 7 line 12) Add comma after "e.g."*

Sentence amended as suggested

*(page 8 line 12) "...instead reflecting more sunlight thus enhancing the albedo..." This*

*clause is confusing. What is it that is reflecting more sunlight?*
We have restructured this sentence to be clearer: 'Over the Antarctic ice sheets, both TOA outgoing and surface incoming SW radiation are overestimated, due to an underestimation of low clouds which allows the high albedo to reflect too much incoming SW radiation back out to space.'

*(page 8 line 20) Replace "...low clouds allowing.." with "...low clouds, which allows. . ."*
Sentence amended as suggested

*(page 8 line 24) "This estimation is slightly greater. . ." Greater than what?*
We have amended this sentence to say greater than the CMIP5 GCMs

*(page 9 line 20) Replace "representing" with "which is due to" or "which represents" to avoid awkward verb-gerund construction ("...representing lofting...").*
Sentence amended as suggested

*(page 9 line 23) Use appropriate subscript formatting for H2SO4.*
Typo amended

*(page 9 line 25) Replace "...in new particle formation, forming secondary sulfate aerosol. . ." with "...in formation of secondary sulfate aerosols,..."*
Sentence amended as suggested

*(page 9 line 26) Correct formatting of H2SO4.*
Typo amended, this has been checked throughout the text

*(page 10 line 5 and throughout manuscript) Replace "Whilest" " with "While"*
This has been amended thought the manuscript as suggested

*(page 10 line 26) Replace "cloud condensation nucleii" with "CCN"*
Sentence amended as suggested

*(page 10 line 26-33) "Fig. 8d-e show the number. . ." The paragraph does not clearly describe the figures; sentence structure is awkward. Consider revision as follows:*

*Fig. 8d-e show the number concentration of CCN with dry diameters greater than 70 nm (CCN70) for the Ctl and the differences resulting from Exp.1. The largest absolute differences are in the tropics, which, similarly to the N3, have the highest concentration. Relatively, there is a global decrease of 5%, whilst decreases of 7% were found over the Australian region, decreases of 8% over the SO and decreases of 20% over the SEP. Differences in cloud droplet number (CDN) are shown in Fig. 8g-h. The relative differences in CDN (Exp.1-Ctl) show a similar spatial pattern to that of the CCN. Global mean CDN decreases by 5%. A decrease of 5% is also found for the Australian region, whereas the SO shows an 8% decrease, and the SEP shows an 18% decrease. In both the CCN70 and CDN, the marine Southern Hemisphere mid-latitudes have the largest decreases of 14% (averaged between 5-35ËŽS) despite the SO having some of the larger decreases in SO2 and H2SO4.*
The figure caption has been amended as suggested by the Reviewer.

*(page 10 line 30) "Exp 1- Ctl" Is this explaining the ratio for relative difference? It is not given elsewhere when talking about relative difference.*
The 'Exp 1- Ctl' notation simply implies Experiment 1 minus the Control. We have removed it to avoid confusion.

*(page 11 line 1) Replace "tropics mid-latitudes" with "mid-latitude tropics".*
Sentence has been amended

*(page 11 line 13) "Similar CCN sensitivities are reported in the Woodhouse et al. (2010) study." State actual values reported in the reference.*
Actual values have been provided

*(page 11 line 24) "...radiation scheme via aerosols and some (see Section 2.1.1)." Missing word/phrase following "some".*
Missing words was 'gases'

*(page 11 line 28) "...slightly removed from the coastline." Not clear. From the Southern Hemisphere's coastline? Recommend revising the whole first sentence, ending with a*

[Figure]

*period after "(Fig. 9a)" and then beginning a new sentence. This would help to clarify what difference is seen in the SH generally and would parallel the structure used for the rest of the paragraph.*
We have revised this sentence and removed the confusing reference to the coastline.

*(page 11 line 32) "...see Section 3 in comparison to other areas. . ." Confusing. The author is comparing all of Section 3 to the areas of low cloud formation?*
A typo has been amended in the sentence (a missing bracket after Section 3) which resolves the clarity of this section.

*(page 11 line 35) "...biases exacerbate this. . ." Clarify what "this" refers to.*
We have clarified what 'this' refers to (the differing responses of areas of low cloud formation).

*(page 12 line 4) Complete parentheses for "shown in Fig. 10a-b"*
Typo amended

*(page 12 line 8) Capitalize "Southern Hemisphere"*
Typo amended

*(page 12 line 13) Add "the" before "Ctl"*
Sentence amended as suggested

*(page 12 line 25) Delete the comma after*
"magnitude" Sentence amended as suggested

*(page 12 line 28) Revise text to "...the SEP: increases of 42%, 172%, and 89% respectively."*
Sentence amended as suggested

*(page 12 line 29) Insert "occurs" after "...a decrease of 37%"*
Sentence amended as suggested

*(page 12 line 33) Insert "respectively" after "...increases by 6%, 4% and 5%"*

Sentence amended as suggested

*(page 12 line 34) Insert "the" before "...SEP of 14%"*
Sentence amended as suggested

*(page 13 line 2) Replace "showing" with "which show"*
Sentence amended as suggested

*(page 13 line 2) Suggest "Incoming surface SW radiation" rather than "Surface incoming SW radiation. . ."*
Sentence amended as suggested

*(page 13 line 6) Replace "analyzed under" with "which analyzed"*
Sentence amended as suggested

*(page 13 line 8) Replace "south east" with "southeast"; Replace "these results presented" with "the results presented here"*
Sentence amended as suggested

*(page 13 line 10) Insert "however" between "...the surface," and "the precipitation. . ."*
Sentence amended as suggested

*(page 13 line 18) Delete "Although"; insert "and" between "...warming" and "we. . ."*
Sentence amended as suggested

*(page 13 line 21) Delete comma after "...study"*
Typo amended

*(page 13 line 26) Replace "on" with "in"*
Typo amended

*(page 13 line 30) Delete duplicate "the"; insert "that" between "...this study is" and "the model. . ."; replace "underestimation of" with "underestimates"*
Sentence amended as suggested

*(page 14 line 1) Replace "rather a multitude of theories" with "multiple theories have been proposed"*
Sentence amended as suggested

*(page 14 line 7) "representation" should be plural*
Typo amended as suggested

*(page 14 line 13) Change "Figure 1" to "Fig. 1" for consistency with rest of manuscript*
Sentence amended as suggested

*(page 14 line 14) Delete comma after SO2*
Sentence amended as suggested

*(page 14 line 32) Break run-on sentence: Period after "importance". Next sentence: "Instead, the studies highlighted SO (Thomas et al., 2010; Mahajan et al., 2015)." Highlighted the SO in terms of what parameter(s)?*
These studies have focused on cloud feedbacks, which we have clarified in the text.

*(page 14 line 35) Revise sentence as: "...slightly lower than the estimation of 2.03 Wm−2 by Thomas. . ."*
Sentence amended as suggested

*(page 15 line 1) Replace "are" with "were"*
Sentence amended as suggested

*(page 15 line 5) Revise reference format: (Albrecht, 1989)*
Sentence amended as suggested

*(page 15 line 16) Revise placement of subordinate clause in the sentence beginning "The cause of the discrepancy. . ." as: "Without further information, it is difficult to speculate on the cause of the discrepancy. . ."*
Sentence amended as suggested

*(page 15 line 18) "Suggest" should be plural; delete comma after constraints*

Sentence amended as suggested

*(page 15 line 22) Replace "Whilst" with "Though"*
Sentence amended as suggested

*(page 15 line 23) Replace the colon after "For example" with a comma; replace "an as of yet" with currently; replace "for" with "as a"*
Sentence amended as suggested

*(page 15 line 31) Replace the semicolon after "Six et al (2013) with "and"*
Sentence amended as suggested

*(page 16 line 3) "coral-reef-derived" (2 hyphens), or "DMS derived from coral reefs"*
Sentence amended as suggested

*(page 16 line 5) Replace semicolon after "Hopkins et al. (2011)" with a comma*
Sentence amended as suggested

*(page 16 line 8) Insert "those" between "than" and "found"*
Sentence amended as suggested

*(page 16 line 12) Make example given in this sentence parenthetical: "...increases (e.g., via solar radiation management) may have a short term cooling effects, however, without. . ."*
Sentence amended as suggested

*(page 16 line 13) Delete the "a" between "...may have" and "short term..."; insert comma after "...however"*
Sentence amended as suggested

*(page 16 line 14) Delete "on" between "...impact" and "marine life. . ."*
Sentence amended as suggested

---

## Author Response (AR2)

**Author Response and Manuscript Revision**

Sonya Fiddes

April 2018

Dear Prof. Takemura,

Please find our response to the Reviewer's comments and our revised manuscript 'Cloud, precipitation and radiation responses to large perturbations in global dimethyl sulfide'. We are grateful for the Reviewer's careful and considered comments and believe that our manuscript has benefited from the close attention provided.

We apologise for not including this cover letter in the previous replies. This may have lead to some confusion about our replies to the inline comments of Reviewer 2, many of which are general and apply throughout the manuscript. We have now highlighted where and how we have made changes in response to all significant comments in the replies. A version of the new manuscript containing the 'tracked changes' (resulting from the latexdiff tool) is also now included.

In particular, both Reviewers commented on the novelty of this study. Whilst we agree that there have been other similar studies, we find that none of them address the question of the role of DMS in the climate system as completely as we do, or with the broad perspectives we do. Specifically, our study considers the complete range of chemical, aerosol, meteorological and climate responses to perturbed DMS fluxes, and discusses these results in the context of climate observational and modelling uncertainty. This is unique in the literature at the present time. Considering these Reviewer comments, and taking suggestions from the Reviewers, have added extra emphasis to this novelty in the manuscript. Furthermore, we have included greater discussion in the introduction and conclusion about the limitations of previous work and continued that discussion throughout the manuscript. Whilst our study does not resolve all of the limitations of the field, we believe, as Reviewer 1 notes, that we address 'current scientific questions, which are not yet fully answered' and as Reviewer 2 notes, these questions are 'timely and relevant'. We have also now provided suggestions as to what directions future studies could take to address these ongoing questions, as suggested by Reviewer 1. We believe that our manuscript, in highlighting these limitations and placing them in a broader perspective, can encourage and provide a starting point for the scientific community to improve our knowledge of this complex and non-linear system.

We hope that you find our responses and revisions satisfactory and that our manuscript is now acceptable for publication in *Atmospheric Chemistry and Physics*.

Kind regards, Sonya Fiddes

**Response to Reviewer 1**

We would like to thank Reviewer 1 for their comments on our manuscript. We agree that while similar simulations have been conducted, our focus has been on providing a complete end-to-end evaluation of the role of DMS in the climate system, from changes in the chemistry through to effects on clouds, precipitation and radiation. Furthermore, we have highlighted the problems remaining when trying to model and understand this complex system. We have made these discussion points clearer in both the introduction and conclusion, which has strengthened the arguments presented in this work. Furthermore, we have clarified our methods, which hopefully leaves no remaining ambiguity. We have addressed the specific comments below, and we hope you find our revisions satisfactory.

**General Comments:**

*In the abstract not all results are summarized, like the comparison of the control simulation with observations, the second experiment with increased DMS emissions, and the estimate of temperature response to changes in DMS TOA radiative effect. The final sentence of the abstract is too general, the authors should try to briefly explain what kind of model study are further needed and why.*
We have revised the abstract to include the model evaluation, results of both experiments and the temperature sensitivity to DMS flux. We have also made the abstract more specific.

*The explanation given at the end of section 2.1.1 about nudging is not very clear. I understand that doing nudged simulation is better to compare the control model simulation with observation. On the other hand the disadvantage is to not have the full impact on meteorology when comparing the control simulation with the two sensitivity experiments. As far as I understand the model simulation includes both direct and indirect effect of aerosols on radiation, so it is not very clear the sentence at page 4, lines 15-18. Maybe also the paragraph at page 14, lines 3-5 should be included in section 2.1.1.*
We have revised our explanation and justification of using nudging in our simulations in the methods section as suggested (page 4, lines 15-21). The reviewer is correct that the model can simulate direct and indirect aerosol effects. Our use of the word 'direct' in this section may have been misleading and we have rectified this. We have also revised the text on the use of nudging in the discussion section (page 15, line 16).

*The quality of the figures is not always satisfying. In particular the figures which include multiple maps are too small and it is difficult to visualize the fields. Also sometime the colors does not help the data visualization. In particular I would recommend to improve Figures 2,3,4,5,6,8 and 10.*
We have adjusted the colour scales by removing the darkest colours to increase visibility of the figures. We have also removed white space where we can and enlarged the figures. We hope these changes make our figures easier to interpret. We will also ensure the figures are readable after the typesetting process.

*The section 5 is too long and somehow difficult to read. I would recommend to split in two or to shorten it by removing some of the details which are repeated from the previous sections. I would try to explain better the last part of the discussion, providing more details on what kind of experiments are needed to better understand the role of DMS future, considering the ocean acidfcation, as mentioned in the last paragraph.*
We have significantly shortened this section, cutting out repetitive paragraphs summarising results previously discussed as suggested by the reviewer. We strengthened our discussion involving not only the questions we are addressing but also pointing towards what steps need to be taken for future studies. We are hesitant to include more information about the impacts of ocean acidification as it is beyond the scope of this study (and admittedly, not our area of expertise). However, we do wish to highlight potential implications of ocean acidification on climate that may not have previously been considered.

**Minor Comments:**

*Page 3, line 1: Six et al. (2013)*
Typo has been amended
*Page 3, Line 21: the control simulation is not explained, only the two DMS perturbation C2 simulations are described*

We have included a description of the control simulation: 'Three simulations are performed to explore the chemical, aerosol and meteorological implications of large $DMS_w$ perturbations. In the first experiment, a control simulation is compared to a simulation in which all $DMS_w$ is removed from the system to determine its current contribution to the climate. In the second experiment, the control simulation is compared to a simulation in which $DMS_w$ is significantly increased, and the results are compared to that of the work by Grandey and Wang(2015)'. More information about the control simulation is provided in Section 2.1 and 2.3.

*Page 4, Line 1: In this study, . . .*
We are unsure what the reviewer meant here. We do not feel starting the paragraph with 'in this study' is appropriate.

*Page 4, Line 10: did you forget CO2 from the list of GHG gases?*
CO2 has now been explicitly mentioned, it is prescribed to the model as global mean concentration: 'Long-lived greenhouse gas concentrations (e.g.$CO_2$,$CH_4$, and $N_2O$) are prescribed from Coupled Model Intercomparison Project Phase 5 (CMIP5) and RCP6.0 recommendations'

*Page 6, Line 1: 50-440 hPa*
Typo has been amended

*Page 6, Line 17-19: WHy these three regions were chosen? A short motivation should be added. I would include the boundaries of these regions in one of the figures.*
We have included motivation as to why these regions have been chosen and included the boundaries in Figure 2: 'Three regions of interest are defined for their relevance to the broader Australian community (for which ACCESS is purposed) or are of particular interest in the DMS-climate system'.

*Page 6, Line 18: Pacific*
Typo has been amended

*Page 6, Line 29: for the first time the CTL name is used, should be introduced before*
The Ctl name has been introduced a few paragraphs above.

*Page 6, Line31: "without the need for an . . .", I would remove or rephrase as an ensemble experiment of free-running climate simulations is needed to better quantify the impact of DMS forcing on the temperature.*
We have rephrased this sentence as suggested 'In ACCESS-UKCA, an ensemble experiment would be required to provide equivalent temperature difference estimates ...'

*Page 7, Line 9: fraction larger then 0.5 instead of ¿*
Typo has been amended

*Page 8, Line 18: outgoing TOA, LW or SW?*
We have clarified this (SW)

*Page 8, Line 30: (Fig 5c)? positive bias if is the difference between the model and observations, not clear from figure caption.*
We have revised the figure caption for clarity: 'The mean (2000-2009) annual total precipitation of a) the satellite climatology from TRMM (Huffman et al., 2007)), b) the ACCESS-UKCA climatology and c) the difference between ACCESS-UKCA and the TRMM product (model - observations). Units are in year$^{-1}$'

*Page 8, Line 30: over regions to the north and south if the equator, .. but only in the tropics is over 2000 mm/yr*
We have revised this sentence for clarity: 'Precipitation in ACCESS-UKCA has large positive biases in regions that receive the most annual rainfall and align with the Intertropical and South Pacific Convergence Zones (ITCZ/SPCZ). These regions overestimate precipitation by over 2000mm year$^{-1}$ '

*Page 9, Line 4: the figure with % differences is not shown, but the values of the largest % differences could be inserted in the text.*

We have included the % differences in the text as suggested: 'the largest differences occur in the eastern basins of the South Pacific (493% over the SEP region) and South Atlantic Oceans (275% from 0 to 25S,330 to 10E).'

*Page 9, Line 17: The largest absolute differences are in the tropics and mid-latitudes over the Oceans.*
Sentence amended as suggested: 'The largest absolute differences are in the tropics and mid-latitudes over the oceans'

*Page 10, Line 5: Fig6 g-h is not correct*
Typo has been amended

*Page 10, Line 21: the largest absolute differences are in clean terrestrial regions. C3 Which regions? Not easy to visualize in the figures. Too small.*
We have removed reference to these differences due to the difficulty of visualisation.

*Page 11, Line 11: the results presented here suggest a lower CCN . . .*
Sentence amended as suggested

*Page 11,Line 18: would title the section "Clouds and precipitation response"*
Title amended as suggested

*Page 11, Line 21-23: this explanation about nudging should be explained also before when describing the experiments.*
We have revised our explanation of nudging in the Methods section (see previous comment).

*Page 11, Line 32: (see section 3) Page 12, Line 4: remove ) after fig 10a-b*
Typo has been amended

*Page 13, Section 4.3: Is it possible to put the uncertainties of these estimates of temperature changes per Tg of S emitted?*
We have performed a moving block bootstrap to determine the uncertainty of the mean changes in radiation, via the 10th and 90th percentile confidence intervals. This can then be translated into a change in temperature by the FAIR model and applied to the flux sensitivity calculations. Further description of how this was performed is now included in the Methods and Tables 3 and 4 have been updated to include these uncertainties.

*Page 14, Line 33: put ref of thomas and mahajan in parenthesis*
Typo has been amended

*Page 16, Line 3: can you better explain the role of coral-reef derived DMS? Why it is important?*
We have included a sentence describing why coral reef derived DMS is important (the fact that it is unaccounted for in current global modelling): 'For example, recent studies have indicated that coral reefs produce significant amounts of DMS, and are an unaccounted for source of sulfur (Hopkins et al., 2016; Swan et al., 2017; Jones et al., 2017)'. Note that this section has been significantly revised and re-sestrucurd as suggested in the general comments.

*Page 16, Line 8: our results imply that a 25% decrease in . . . would result in an increase of 0.1C. Is it possible to put the uncertainty on this estimate?*
We have now included confidence intervals for our temperature/flux sensitivity calculations, and the estimates of potential temperature change with decreasing DMS flux.

*Caption of figure 3: third column is not absolute differences, as negative numbers are shown.*
Caption has been amended

*Caption Figure 5: c) not clear if difference model-obs or contrary while reading the description of the figure in the manuscript.*
We have revised the caption for clarity (see previous comment)

*Caption Figure 7: blue lines show the SO (Southern Ocean) mean, red the australian (aus). The short name aus is used only here and not in the manuscript.*

We have only shortened Australia to Aus. for clarity in this figure.

**Response to Reviewer 2**

We would like to thank Reviewer 2 for their extensive comments.

We acknowledge that there are a number of recent studies examining the role of DMS in the climate system. Our study provides a thorough end-to-end analysis of all aspects of the chemical, aerosol and meteorological impacts of large DMS perturbations in the current climate. To our knowledge, previous studies have not considered the entire system, and typically only considered short time periods (1yr) or future projections. We have made these points clear in the Introduction (page 4, line 1-13), and more specifically noted this on page 4, line 10-13: 'Many of the studies noted above focused on one or two aspects of the DMS-climate system, commonly reporting on the flux$_{DMS}$ and its radiative and temperature effects. In this study we evaluate the whole system, examining chemical, aerosol and meteorological changes, including cloud and precipitation effects'

We would also like to thank Reviewer 2 for bringing to our attention the two Tesdal et al. 2016 papers, in addition to the the other papers listed in the point by point comments. They have provided a useful reference for discussion around DMS climatology and flux uncertainties. We have included the references suggested by the Reviewer where appropriate, as listed in the point by point comments.

We have taken the Reviewer's advice and have strengthened not only our problem statement in the Introduction (in particular on Page 4 line 18-20: 'We aim to discuss these sensitivities not only within the specific context of the DMS-climate system, as mentioned above, over a 10 year time period, but also in the broader context of the current uncertainties in the DMS-climate system and climate modelling') but also the points mentioned above in the Introduction, which help to better identify gaps in the literature and frame our study.

We agree that just using a different model does not provide sufficient novelty to a study and it was not our intention for this to be a main aim of the paper. By clarifying our problem statement above, we have highlighted the true aim and novelty of this study. ACCESS-UKCA is an established and respected global climate-chemistry model, and for this reason we have not provided any comment comparing ACCESS-UKCA to other global chemistry-climate models. Nevertheless, we have included some more information in the Methods about why ACCESS-UKCA, but more particularly GLOMAP-mode, is a desirable tool (including references suggested by this Reviewer, Page 5, line 31-33: 'A full description of the scheme can be found in Mann et al. (2010) with improvements detailed in Mann et al. (2012). Bellouin et al. (2013) compare GLOMAP-mode with an older generation aerosol scheme, finding significant differences in aerosol response to perturbations between the two schemes'), and further elucidated the novel aspects of the study in the manuscript.

We also agree that the experiments performed here are extreme cases. However, there are significant insights into the DMS-climate system gained from these experiments that will serve to inform more targeted, realistic scenario experiments. Specifically, our large scale perturbations allow quantification of the overall contribution of DMS to the climate system (Section 4.1, page 11). More subtly, this study has identified non-linearities in the DMS-climate system, highlights areas of particular sensitivity , provided evidence of important cloud-aerosol processes and has placed these results in a broad context of our current understanding of the DMS-climate system and the associated uncertainties. We have enhanced these results throughout the Conclusion (in particular the paragraphs, page 16 line 3-20).

**Point by point comments:**

*The introduction provides relevant background regarding the role of aerosols in the global radiation budget. It would benefit from more background information regarding the uncertainty in DMS concentrations in*

*the ocean. See for example: Belviso et al. (2004), Tesdal et al. (2016a).*
We have included further background information regarding the uncertainties surrounding the DMS climatology and flux as suggested by the Reviewer (see paragraph page 2, line 29 - page 3, line 3).

*(page 2 line 9): Additional support for the use of DMS fields in climate model are given in Belviso et al. (2004) and Tesdal et al. (2016a), and it is recommended to include these references. Hopkins et al. (2016) is not an appropriate reference for the uncertainty around observed DMS concentrations. Royer et al. (2015) is suggested.*
We have revised this section, removed parts and included more appropriate references. It now reads as 'Global DMS concentrations and fluxes remain poorly constrained by observations (Tesdal et al., 2016; Royer et al., 2015), ... '

*(page 2 line 10) Charlson et al (1987) is not an appropriate reference for the role of DMS in climate systems being subject to debate, as this is the paper that introduced the hypothesis. The debate came later. A more appropriate article that also serves as an review of the CLAW hypothesis is Ayers and Cainey (2007).*
We have included the recommended citation as the Reviewer has suggested: '... its role in the climate system is subject to debate (Ayers and Cainey, 2007; Quinn and Bates, 2011)'.

*(page 2 line 16) Add "s" to "contribute"*
Typo amended

*(page 2 line 23) Add "in certain regions" following "...local DMS concentrations"*
Sentence amended as suggested

*(page 2 line 29) Add "in order" between "... a one year simulation" and "to quantify its importance. . ."*
Sentence amended as suggested

*(page 2 line 32) Replace "for example" with "e.g."*
Sentence amended as suggested

*(page 2 line 34) Is ocean acidification meant to be an example of "anthropogenic climate change"? Please clarify.*
This sentence has been clarified to reflect that ocean acidification is caused by anthropogenic emission of CO2. '... found DMS emissions were reduced by 17% by the end of the century, primarily due to decreasing ocean pH (caused by anthropogenic CO2 emissions)'.

*(page 3 line 3 and throughout manuscript) Both Celsius and Kelvin temperature units are used throughout. Please use one or the other (preferably Celsius) for consistency.*
We have made all temperature units consistent throughout the text

*(page 3 line 10) Replace "low-mid level" with "low- and mid-level"*
Sentence amended as suggested

*(page 3 line 26) Add "analyzes" after "Section 4"*
Sentence amended as suggested

*(page 4 line 23) Replace "of" with "detailed in"*
Sentence amended as suggested

*(page 4 line 28) " A full description of the scheme can be found in Mann et al. . ." Here it would be appropriate to mention the study by Bellouin et al. (2013), who compared GLOMAP-mode and CLASSIC and determine that GLOMAP-mode is more sophisticated and accurate.*
We have made reference to the Bellouin et al. (2013) paper, as well as the Mann et al (2012) paper. We have retained the Mann et al. (2010) citation as this is the model description paper. This now reads as: ' A full description of the scheme can be found in Mann et al. (2010) with improvements detailed in Mann et al. (2012). Bellouin et al. (2013) compare GLOMAP-mode with an older generation aerosol

scheme, finding significant differences in aerosol response to perturbations between the two schemes'

*(page 5 line 1) "Significant sampling biases...Northern Hemisphere (Lana et al 2011)" This sentence is not entirely clear. Is the point that the data is biased towards springsummer and towards Northern Hemisphere?*
Yes, we have clarified this sentence to reflect this: 'Significant sampling biases exist within the Lana et al. (2011) data set, with approximately half of observations collected in late spring through summer, and more than two-thirds of the data collected in the Northern Hemisphere.'

*(page 5 line 2) Replace "spring-summer" with "spring through summer"*
Sentence amended as suggested

*(page 5 line 5) "The Liss and Merlivat (1986) parameterization. . ." There should be justification for why this parameterization is used. Several different parameterizations of the piston velocity in terms of wind speed have been used in modelling studies (e.g., Liss and Merlivat, 1986; Wanninkhof, 1992; Nightingale et al., 2000), leading to substantially different flux fields for a given concentration field (Tesdal et al., 2016a).*
We have made reference to the alternative flux parameterisations as well as the modelling studies that detail their uncertainties in the Introduction (page 2, line 29 - page 3, line 3). Of the parameterisations available within ACCESS-UKCA, we chose the Liss  Merlivat (1986) method because it agrees more closely with the latest generation of DMS flux parameterisations, e.g. Vlahos  Monahan (2009), Bell et al. (2017), which were specifically derived for DMS, unlike the above parameterisations. We have updated the text to reflect this: 'The most common flux parameterisations exhibit considerable ranges in $flux_{DMS}, from 15-35 Tg year^1$ of sulfur in Elliott (2009) to 9-34 Tg year[1] of sulfur found in Tesdal et al. (2016), who recommend a range of 18-24 Tg year[1] of sulfur as a reasonable estimate. Vlahos and Monahan (2009) and Bell et al. (2017) show that current parameterisations overestimate the $flux_{DMS}$ at high wind speeds and suggest that annual global $flux_{DMS}$ is likely to be in the lower range of current estimates. Of the $flux_{DMS}$ parameterisations available in ACCESS-UKCA, the Liss and Merlivat (1986) method yields a low to moderate flux comparable to those calculated in Vlahos and Monahan (2009) and Bell et al. (2017), and is used in this study.'

*(page 5 line 16-17) Following Eq. 4, Replace "Where" with "Here," . Insert "is" before "determined following the method of Saltzman.."*
Sentence amended as suggested

*(page 5 line 20) "seawater" should not be*
hyphenated Typo amended

*(Section 2.2 Model Evaluation) The evaluation should include comparison to observation of atmospheric concentration of DMS and other sulfur species, for example as described in Tesdal et al. (2016b).*
We appreciate that an evaluation of atmospheric DMS and other sulfur species is desirable, as done in the Tesdal et al. (2016b) study. However, we have limited this evaluation to meteorological data sets that are globally available from satellite products.We are confident in the use of this model for DMS studies due GLOMAP-mode being extensively evaluated against observations in many previous studies (Mann et al. 2010, 2012, Woodhouse et al 2010, 2013, 2015).

*(page 6 line 1) add "between" between "medium" and "440-680 hPa" and between "low" and "680-1000 hPa"*
Sentence amended as suggested

*(page 6 line 17) Add period at end of sentence that begins "The Australian region...." (page 6 line 24) Replace "that" with "the one"*
Sentence amended as suggested

*(page 6 line 30) "By providing this radiative effect. . ." The text appears to imply that FAIR is a tool that provides estimates of climate response, given a simple value of radiative effect. Thus, by feeding estimates of radiative effects to FAIR one can analyze the effect on temperature and other climate variables.The text would benefit from more elaboration regarding the relationship between the radiative effect*

*and the FAIR climate component.*
We have included some more information on how the FAIR emulator links radiative forcing changes to temperature and we have included some additional references for readers to pursue. This section now reads as: 'FAIR's climate component is a simple impulse response model which emulates the behaviour of more complex Earth System Models, given a certain radiative forcing (in this case due to DMS). FAIR has been designed to determine temperature responses to radiative forcing of similar magnitudes to the DMS radiative effect (Millar et al., 2017). FAIR's temperature response is calculated as the sum of two components, approximately10representing the response of the upper mixed layer and deep ocean to a change in radiative forcing (Millar et al., 2017). Due to its simplicity, FAIR cannot capture the non-linearities and feedbacks in the climate system, and hence the temperature response calculated must be taken as an estimate only. Furthermore, in this work we consider only a single mid-range estimate of climate sensitivity'

*(page 7 line 12) Add comma after "e.g."*
Sentence amended as suggested

*(page 8 line 12) "...instead reflecting more sunlight thus enhancing the albedo..." This clause is confusing. What is it that is reflecting more sunlight?*
We have restructured this sentence to be clearer: 'Over the Antarctic ice sheets, both TOA outgoing and surface incoming SW radiation are overestimated, due to an underestimation of low clouds which allows the high albedo to reflect too much incoming SW radiation back out to space.'

*(page 8 line 20) Replace "...low clouds allowing.." with "...low clouds, which allows. . ."*
Sentence amended as suggested

*(page 8 line 24) "This estimation is slightly greater. . ." Greater than what?*
We have amended this sentence to say 'greater than that found for CMIP5 GCMs'

*(page 9 line 20) Replace "representing" with "which is due to" or "which represents" to avoid awkward verb-gerund construction ("...representing lofting...").*
Sentence amended as suggested

*(page 9 line 23) Use appropriate subscript formatting for H2SO4.*
Typo amended

*(page 9 line 25) Replace "...in new particle formation, forming secondary sulfate aerosol. . ." with "...in formation of secondary sulfate aerosols,..."*
Sentence amended as suggested

*(page 9 line 26) Correct formatting of H2SO4.*
Typo amended, this has been checked throughout the text

*(page 10 line 5 and throughout manuscript) Replace "Whilest" " with "While"*
This has been amended thought the manuscript as suggested

*(page 10 line 26) Replace "cloud condensation nucleii" with "CCN"*
Sentence amended as suggested

*(page 10 line 26-33) "Fig. 8d-e show the number. . ." The paragraph does not clearly describe the figures; sentence structure is awkward. Consider revision as follows: Fig. 8d-e show the number concentration of CCN with dry diameters greater than 70 nm (CCN70) for the Ctl and the differences resulting from Exp.1. The largest absolute differences are in the tropics, which, similarly to the N3, have the highest concentration. Relatively, there is a global decrease of 5%, whilst decreases of 7% were found over the Australian region, decreases of 8% over the SO and decreases of 20% over the SEP. Differences in cloud droplet number (CDN) are shown in Fig. 8g-h. The relative differences in CDN (Exp.1-Ctl) show a similar spatial pattern to that of the CCN. Global mean CDN decreases by 5%. A decrease of 5% is also found for the Australian region, whereas the SO shows an 8% decrease, and the SEP shows an 18% decrease. In both the CCN70 and CDN, the marine Southern Hemisphere mid-latitudes have the largest*

decreases of 14% (averaged between 5-35ĔŽS) despite the SO having some of the larger decreases in SO2 and H2SO4.
The paragraph has been amended as suggested by the Reviewer.

*(page 10 line 30) "Exp 1- Ctl" Is this explaining the ratio for relative difference? It is not given elsewhere when talking about relative difference.*
The 'Exp 1- Ctl' notation simply implies Experiment 1 minus the Control. We have removed it to avoid confusion.

*(page 11 line 1) Replace "tropics mid-latitudes" with "mid-latitude tropics".*
Sentence has been amended

*(page 11 line 13) "Similar CCN sensitivities are reported in the Woodhouse et al. (2010) study." State actual values reported in the reference.*
Actual values have been provided 'Similar CCN sensitivities are reported in the Woodhouse et al. (2010) study (63cm3/mg m$^2$ day$^1$ global average).'

*(page 11 line 24) "...radiation scheme via aerosols and some (see Section 2.1.1)." Missing word/phrase following "some".*
Missing words was 'gases'

*(page 11 line 28) "...slightly removed from the coastline." Not clear. From the Southern Hemisphere's coastline? Recommend revising the whole first sentence, ending with a period after "(Fig. 9a)" and then beginning a new sentence. This would help to clarify what difference is seen in the SH generally and would parallel the structure used for the rest of the paragraph.*
We have revised this sentence and removed the confusing reference to the coastline: 'The largest differences occur in eastern basins of the Southern Hemisphere's oceans. '

*(page 11 line 32) "...see Section 3 in comparison to other areas. . ." Confusing. The author is comparing all of Section 3 to the areas of low cloud formation?*
A typo has been amended in the sentence (a missing bracket after Section 3) which resolves the clarity of this section.

*(page 11 line 35) "...biases exacerbate this. . ." Clarify what "this" refers to.*
We have clarified what 'this' refers to (the differing responses of areas of low cloud formation): '... whether the long standing model biases, especially those around the formation of supercooled liquid water,have contributed to the differing responses requires further investigation'

*(page 12 line 4) Complete parentheses for "shown in Fig. 10a-b"*
Typo amended

*(page 12 line 8) Capitalize "Southern Hemisphere"*
Typo amended

*(page 12 line 13) Add "the" before "Ctl"*
Sentence amended as suggested

*(page 12 line 25) Delete the comma after "magnitude"*
Sentence amended as suggested

*(page 12 line 28) Revise text to "...the SEP: increases of 42%, 172%, and 89% respectively."*
Sentence amended as suggested

*(page 12 line 29) Insert "occurs" after "...a decrease of 37%"*
Sentence amended as suggested

*(page 12 line 33) Insert "respectively" after "...increases by 6%, 4% and 5%"*
Sentence amended as suggested

*(page 12 line 34) Insert "the" before "...SEP of 14%"*
Sentence amended as suggested

*(page 13 line 2) Replace "showing" with "which show"*
Sentence amended as suggested

*(page 13 line 2) Suggest "Incoming surface SW radiation" rather than "Surface incoming SW radiation. . ."*
Sentence amended as suggested

*(page 13 line 6) Replace "analyzed under" with "which analyzed"*
Sentence amended as suggested

*(page 13 line 8) Replace "south east" with "southeast"; Replace "these results presented" with "the results presented here"*
Sentence amended as suggested

*(page 13 line 10) Insert "however" between "...the surface," and "the precipitation. . ."*
Sentence amended as suggested

*(page 13 line 18) Delete "Although"; insert "and" between "...warming" and "we. . ."*
Sentence amended as suggested

*(page 13 line 21) Delete comma after "...study"*
Typo amended

*(page 13 line 26) Replace "on" with "in"*
Typo amended

*(page 13 line 30) Delete duplicate "the"; insert "that" between "...this study is" and "the model. . ."; replace "underestimation of" with "underestimates"*
Sentence amended as suggested

*(page 14 line 1) Replace "rather a multitude of theories" with "multiple theories have been proposed"*
Sentence amended as suggested

*(page 14 line 7) "representation" should be plural*
Typo amended as suggested

*(page 14 line 13) Change "Figure 1" to "Fig. 1" for consistency with rest of manuscript*
Sentence amended as suggested

*(page 14 line 14) Delete comma after SO2*
Sentence amended as suggested

*(page 14 line 32) Break run-on sentence: Period after "importance". Next sentence: "Instead, the studies highlighted SO (Thomas et al., 2010; Mahajan et al., 2015)." Highlighted the SO in terms of what parameter(s)?*
These studies have focused on cloud feedbacks, which we have clarified in the text: '... instead, these studies focused on cloud feedbacks in the SO'

*(page 14 line 35) Revise sentence as: "...slightly lower than the estimation of 2.03 Wm2 by Thomas. . ."*
Sentence amended as suggested

*(page 15 line 1) Replace "are" with "were"*
Sentence amended as suggested

*(page 15 line 5) Revise reference format: (Albrecht, 1989)*
Sentence amended as suggested

*(page 15 line 16) Revise placement of subordinate clause in the sentence beginning "The cause of the discrepancy. . ." as: "Without further information, it is difficult to speculate on the cause of the discrepancy. . ."*
Sentence amended as suggested

*(page 15 line 18) "Suggest" should be plural; delete comma after constraints*
Sentence amended as suggested

*(page 15 line 22) Replace "Whilst" with "Though"*
Sentence amended as suggested

*(page 15 line 23) Replace the colon after "For example" with a comma; replace "an as of yet" with currently; replace "for" with "as a"*
Sentence amended as suggested

*(page 15 line 31) Replace the semicolon after "Six et al (2013) with "and"*
Sentence amended as suggested

*(page 16 line 3) "coral-reef-derived" (2 hyphens), or "DMS derived from coral reefs"*
Sentence amended as suggested

*(page 16 line 5) Replace semicolon after "Hopkins et al. (2011)" with a comma*
Sentence amended as suggested

*(page 16 line 8) Insert "those" between "than" and "found"*
Sentence amended as suggested

*(page 16 line 12) Make example given in this sentence parenthetical: "...increases (e.g., via solar radiation management) may have a short term cooling effects, however, without. . ."*
Sentence amended as suggested

*(page 16 line 13) Delete the "a" between "...may have" and "short term..."; insert comma after "...however"*
Sentence amended as suggested

*(page 16 line 14) Delete "on" between "...impact" and "marine life. . ."*
Sentence amended as suggested

[revised manuscript text omitted]

$$\text{DMS}_{\text{flux}} = k(\text{DMS}_{\text{w}} - \frac{\text{DMS}_{\text{a}}}{\alpha}) = k(\text{DMS}_{\text{w}}\alpha - \text{DMS}_{\text{a}}) \tag{1}$$

For $w_{10} < 3.6 \ ms^{-1}$:

$$k = 0.17 w_{10} (\frac{SC_{\text{DMS}}}{600})^{\frac{2}{3}} \tag{2}$$

For $3.6 \ ms^{-1} < w_{10} < 13 \ ms^{-1}$:

$$k = 2.85(w_{10} - 3.6)(\frac{SC_{\text{DMS}}}{600})^{\frac{1}{2}} + 0.612(\frac{SC_{\text{DMS}}}{600})^{\frac{2}{3}} \tag{3}$$

For $w_{10} > 13 \ ms^{-1}$:

$$k = 5.9(w_{10} - 13)(\frac{SC_{\text{DMS}}}{600})^{\frac{1}{2}} + 26.79(w_{10} - 3.6)(\frac{SC_{\text{DMS}}}{600})^{\frac{1}{2}} + 0.612(\frac{SC_{\text{DMS}}}{600})^{\frac{2}{3}} \tag{4}$$

[revised manuscript text omitted]